# Leaf-level coordination principles propagate to the ecosystem scale

Ulisse Gomarasca [1] ✉, Mirco Migliavacca[2], Jens Kattge [1], Jacob A. Nelson [1], Ülo Niinemets[3], Christian Wirth [1,4,5], Alessandro Cescatti[2], Michael Bahn [6], Richard Nair [1,7], Alicia T. R. Acosta [8], M. Altaf Arain [9], Mirela Beloiu [10], T. Andrew Black [11], Hans Henrik Bruun [12], Solveig Franziska Bucher [5,13], Nina Buchmann [14], Chaeho Byun [15], Arnaud Carrara [16], Adriano Conte [17], Ana C. da Silva [18], Gregory Duveiller [1], Silvano Fares [19], Andreas Ibrom [20], Alexander Knohl [21], Benjamin Komac[22], Jean-Marc Limousin[23], Christopher H. Lusk[24], Miguel D. Mahecha [5,25], David Martini[1], Vanessa Minden[26], Leonardo Montagnani [27], Akira S. Mori [28], Yusuke Onoda[29], Josep Peñuelas [30,31], Oscar Perez-Priego [32], Peter Poschlod[33], Thomas L. Powell[34], Peter B. Reich [35,36,37], Ladislav Šigut [38], Peter M. van Bodegom [39], Sophia Walther [1], Georg Wohlfahrt [6], Ian J. Wright [37,40] & Markus Reichstein [1,5]

Fundamental axes of variation in plant traits result from trade-offs between costs and benefits of resource-use strategies at the leaf scale. However, it is unclear whether similar trade-offs propagate to the ecosystem level. Here, we test whether trait correlation patterns predicted by three well-known leaf- and plant-level coordination theories – the leaf economics spectrum, the global spectrum of plant form and function, and the least-cost hypothesis – are also observed between community mean traits and ecosystem processes. We combined ecosystem functional properties from FLUXNET sites, vegetation properties, and community mean plant traits into three corresponding principal component analyses. We find that the leaf economics spectrum (90 sites), the global spectrum of plant form and function (89 sites), and the least-cost hypothesis (82 sites) all propagate at the ecosystem level. However, we also find evidence of additional scale-emergent properties. Evaluating the coordination of ecosystem functional properties may aid the development of more realistic global dynamic vegetation models with critical empirical data, reducing the uncertainty of climate change projections.

Decades of research have identified trade-offs and coordination between functional traits at the plant and organ levels that are explained through the concept of eco-evolutionary optimality[1–8]. Optimality assumes that natural selection and environmental filtering shape predictable and general patterns in traits, leading to specific trait combinations that favor the economic efficiency of processes as a necessary condition of plant growth, survival, and reproduction[9]. For instance, the leaf economics spectrum uncovers plant resource harvesting strategies, with underlying trade-offs in the investment and utilization of resources depending on leaf longevity[8]. High structural investments in leaves (high leaf mass per area) translate to slow but long-term carbon gain (high leaf

longevity), while the inverse, mutually exclusive strategy is characterized by high nutrient investments (low leaf mass per area, high leaf nitrogen content per leaf mass) that compensate for short leaf lifespan through increased leaf-level productivity[5,8]. We refer to this as the performance-persistence trade-off, because resource acquisition costs can either be directed toward resource conservation and leaf persistence[10], or fast growth and photosynthetic performance. The global spectrum of plant form and function explores evolutionary strategies related to plant growth, survival, and reproduction by describing two key dimensions related to the size of whole plants and organs, and the performance-persistence trade-off related to the leaf economics spectrum[1]. Another example of trait coordination is the least-cost hypothesis, which describes a continuum in plant economic strategies aimed at optimizing the input mix of two or more key limiting resources. The same economic theory can be applied to resource acquisition and utilization in plants: a decreasing acquisition and retention cost of one of two limited resources (e.g., water) is generally accompanied by an increased cost of the other limiting resource (e.g., nitrogen)[7,11]. While optimality principles and patterns in trait coordination have been widely studied and confirmed at the leaf and plant scale, and some studies at the community scale exist[12], it is unclear how these relationships translate to the ecosystem scale.

Ecosystems are intricate mixtures of different species that compete for resources such as energy, water, and nutrients[13], and abiotic drivers affect biological processes and ecological interactions. Ecosystem-level processes are intrinsically linked to canopy architecture (arrangement of leaves, shoots, etc.)[14,15], and are determined by species composition, but are also influenced by disturbance and management. Consequently, ecosystems feature scale-emergent properties[16–18], i.e., properties that are only manifested at a certain scale. For instance, light interception is largely dependent on canopy architecture due to the amount of light that can penetrate the canopy space[19,20]: whereas light-use efficiency responses observed at the leaf level depend on rather homogenous small-scale conditions, complex gradients of light penetration and light-use efficiencies need to be considered at the canopy scale[21]. In essence, the coordination between ecosystem functional properties at the canopy scale can contrast with the theory of optimization in leaves or plant organs.

Understanding the coordination among functional properties within ecosystems has major implications for the refinement of parameterization and evaluation of terrestrial biosphere models. Several ongoing initiatives are proposing more realism in the coordination of plant functional traits[9,22]. For more realistic predictions of how ecosystems will respond to global environmental changes, the upscaling from leaf or plant to ecosystem-level processes needs to consider whether the coordination observed at leaf- and plant-level is conserved at the ecosystem scale, or, conversely, whether scale-emergent behaviors occur and need to be explicitly implemented in the models[21]. Evidence for ecosystem-level coordination would support the upscaling from simulated leaf-level processes to the ecosystem scale. Concurrently, insight into scale-emergent properties could improve the upscaling algorithms used in dynamic vegetation models and serve to validate the functional response from models.

Here, we ask whether well-established coordination principles that apply to the leaf and plant scales can be used to approximate ecosystem-scale coordination among community mean traits and ecosystem processes. Based on an extensive dataset from 98 global eddy covariance flux measurement sites, and vegetation data collected in-situ and from global databases of plant traits, we explore ecosystem-scale analogs to the relationships between functional traits identified by (i) the leaf economics spectrum[5,8], (ii) the global spectrum of plant form and function[1], and (iii) the least-cost hypothesis[7,11].

## Results and discussion

### Leaf economics spectrum at the ecosystem scale

To analyze whether the leaf economics spectrum[8] propagates to the ecosystem scale, we conducted a principal component (PC) analysis based on five ecosystem functional properties and vegetation properties analogous to the leaf scale (i.e., representing the same or similar process). In our results, each variable is represented by eigenvectors that show their direction and strength in the hyperspace between PCs (Fig. 1a). As in the leaf economic spectrum, we identified the key dimensions, or PCs, that explain the most variance in the data (Fig. 1b). Then, we assessed the projections of the eigenvectors (i.e., loadings) on the PCs and the relative contribution of each variable in defining each PC. We used the community-weighted means of nitrogen per mass (wNmass), leaf longevity (wLL), leaf mass per area (wLMA), the photosynthetic capacity of the whole ecosystem (GPPsat), and the maximum ecosystem respiration (RECOmax, Supplementary Table 1). Two retained PCs cumulatively explained $82.3 \pm 4.7\%$ of the variance in the dataset (Fig. 1). The ecosystem-scale economics spectrum was apparent from the loadings of the PC analysis (PCA) (Fig. 1c). In particular, the first PC showed strong negative loadings of the community-weighted means of leaf mass per area and leaf longevity (wLMA: $-0.83 \pm 0.04$, wLL: $-0.67 \pm 0.06$), and positive loadings of nitrogen content, photosynthetic capacity, and respiration (wNmass: $0.85 \pm 0.03$, GPPsat: $0.79 \pm 0.04$, and RECOmax: $0.69 \pm 0.07$, Fig. 1d and Supplementary Data 1). This greatly substantiates the trade-offs between performance and persistence of the leaf economics spectrum at the ecosystem scale. On the second PC, all variables other than wNmass loaded positively, highlighting scale-emergent positive associations between respiration, photosynthetic capacity, and leaf longevity. Plant functional types (based on the IGBP classification) differed strongly along the axis expressed by the community-weighted mean plant traits. In contrast, the variation within plant functional types was better described by the direction of the GPPsat and RECOmax eigenvectors, with the two sets of variables being nearly orthogonal to one another (Fig. 1a).

Restricting the analysis to forest sites, or evergreen needleleaf forest sites, produced similar results on PC1 as for the overall case with all sites. This hints at the importance of the leaf economics spectrum both within and across plant functional types (Supplementary Fig. 1, Supplementary Data 1).

Results of multi-model inference with different explanatory variables for GPPsat showed higher importance of wNmass compared to wNarea and better model performance when including wNmass and wLMA compared to wNarea, which is one of the reasons why we used wNmass in the PCA (Supplementary Fig. 2).

The results of our first analysis show that the most important dimension of ecosystem functional properties describes the trade-off between performance (productivity) and persistence. This reflects the relationships described in the leaf economics spectrum[8]. At the high productivity side of the spectrum, sites characterized by high photosynthetic capacity, ecosystem respiration, and leaf nitrogen concentration are generally associated with low structural investments for single leaves in the form of low leaf mass per area (i.e., leaf thickness and/or leaf density) and leaf longevity. Low leaf longevity translates to a faster leaf turnover, i.e., possibly higher overall nutrient investments throughout the lifespan of the plant. In contrast, low photosynthetic capacity and respiration rates are associated with lower nitrogen content, extended leaf longevity, and increased leaf thickness/density (i.e., higher wLMA, Fig. 1, Supplementary Data 1).

Osnas et al.[23] criticized the original formulation of the leaf economics spectrum based on mass-normalized traits and leaf mass per area. Here, we used mass-based traits to be coherent with the leaf economics spectrum. Based on the results of the relative importance analysis (Supplementary Fig. 2), we argue that mass-based estimates of nitrogen might be better suited for analyses on ecosystem-level

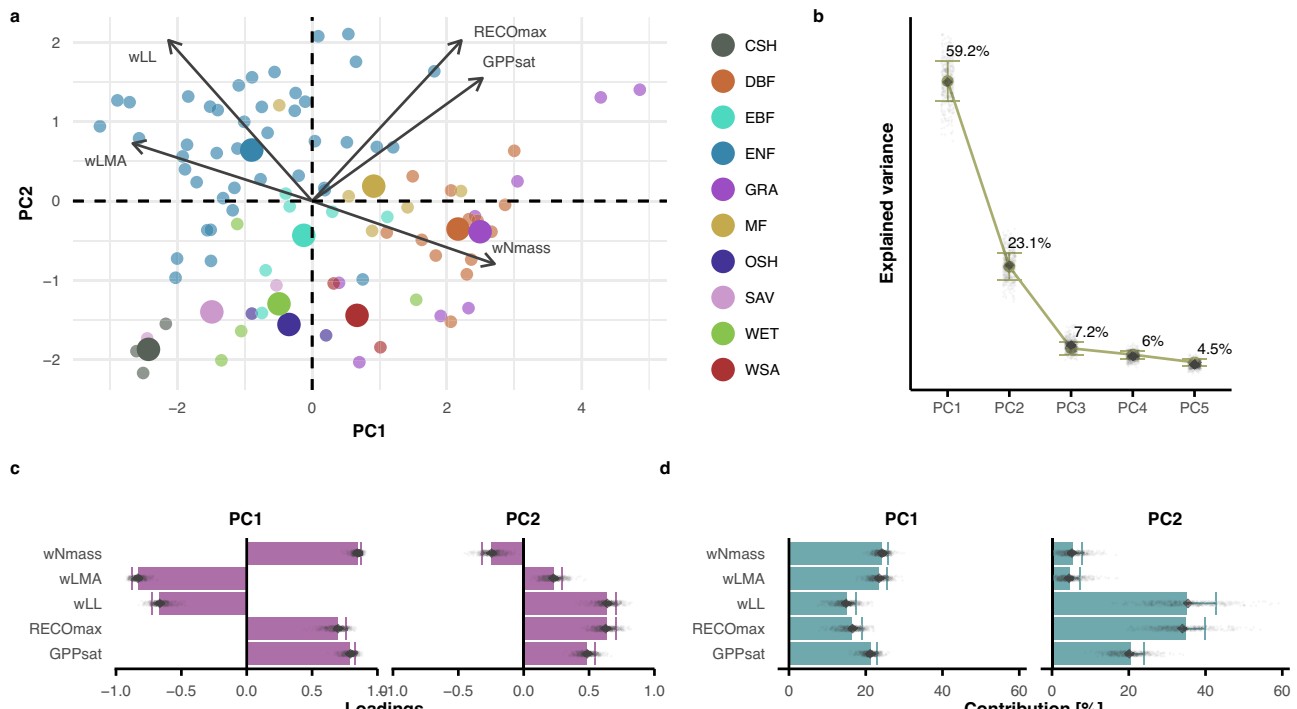

**Fig. 1 | Principal component analysis on variables representing the leaf economics spectrum at the ecosystem scale (90 sites). a** Biplot resulting from PCA; point colors represent plant functional types following the IGBP classification: CSH (Closed Shrubland), DBF (Deciduous Broadleaf Forest), EBF (Evergreen Broadleaf Forest), ENF (Evergreen Needleleaf Forest), GRA (Grassland), MF (Mixed Forest), OSH (Open Shrubland), SAV (Savannah), WET (Wetland), WSA (Woody Savannah). Bigger points represent the centroid of the distribution for each habitat type. **b** Explained variance for the retained principal components (PCs). **c** Barplot for the loadings, and **d** contributions for each variable on the retained PCs. The full circles

in **b** and the bars in **c**, **d** show the pertinent estimate based on the full dataset. In **b**–**d**, the error bars are centered on the estimates and represent the standard error estimated with the bootstrap procedure ($n = 499$ bootstrap iterations); the small gray diamonds show the estimates of each bootstrap iteration, and the big gray diamonds represent the median of all bootstrap iteration. Variable acronyms: gross primary productivity at light saturation (GPPsat), maximum ecosystem respiration (RECOmax), community-weighted mean leaf longevity (wLL), community-weighted mean leaf mass per area (wLMA), community-weighted mean nitrogen per leaf mass (wNmass).

processes, which is in line with previous studies[24]. We also tested the same concept with area-based nitrogen estimates, and we observed very similar results (Supplementary Fig. 3, Supplementary Data 1).

While in the leaf economics spectrum described by Wright et al.[8], the first component explains up to 74% of the variability of the data, the analogous axis at the ecosystem scale explains a lower proportion of variance ($59.2 ± 3.9\%$). However, the additional information shown on the second component at the ecosystem scale adds up to a higher overall explained variance ($82.3 ± 4.7\%$) and suggests higher complexity at the ecosystem scale. The second dimension likely represents scale-emergent properties (i.e., only found at the ecosystem scale) that are not evident with the limited set of variables analogous to the leaf economics spectrum. Finally, we show that additional dimensions are important at the ecosystem scale. In the following sections, we investigate the possibility that this second component is connected with secondary coordination principles (i.e., vegetation size axis of the global spectrum, or least-cost hypothesis component).

## Ecosystem global spectrum of plant form and function

We investigated the role of additional properties related to ecosystem structure by testing, when available, a set of variables analog to the global spectrum of plant form and function[1] at the ecosystem scale. In addition to the variables characterizing the ecosystem-scale economics spectrum (wNmass, wLMA, GPPsat), we included community-weighted stem specific density (wSSD), maximum leaf area index (LAImax), and canopy height (Hc, Supplementary Table 1). Six significant PCs were retained based on the Dray method[25], (Supplementary Data 2). However, we concentrate our interpretation on the first three PCs, as the limited number of sites ($n = 89$) undermines our

capacity to disentangle a large number of dimensions from potential noise in the data. These three components cumulatively explained $82.7 ± 4.3\%$ of the variance in the dataset (Fig. 2).

The first component reflected properties related to the maximum rates of ecosystem processes. Photosynthetic capacity was the main variable contributing positively to PC1, with further strong contributions and positive loadings from all other variables with the exception of wSSD and wLMA (Fig. 2c and Fig. 2d). The only strong negative loading on the first component was the leaf mass per area (wLMA, $-0.61 ± 0.11$). This reflects the performance-persistence trade-off expected by the leaf economics spectrum, with a clear trade-off between process rates and nutrient investments aimed at maximizing productivity, and properties related to long-lived strategies. This shows that the leaf economics spectrum dominates the variability among ecosystem functional properties. The second PC was primarily defined by variables connected to structure and/or foliar chemistry: nitrogen content and stem-specific density loaded negatively (wNmass: $-0.66 ± 0.58$, wSSD: $-0.52 ± 0.56$), while leaf area index and leaf mass per area had strong positive loadings on PC2 (LAImax: $0.60 ± 0.50$, and wLMA: $0.44 ± 0.37$). This second component resembled the size axis of the global spectrum of plant form and function, with structural properties related to total leaf area in the canopy and canopy height. Together, these two axes generate a plane that is strikingly similar to the one described in the study by Díaz et al.[1], confirming the hypothesis that the global spectrum propagates to the ecosystem scale. However, when looking at the direction of the eigenvectors relative to one another, GPPsat falls between the axis of leaf economics (wNmass, wLMA), and the axis of size (LAImax specifically). This suggests that the leaf economics spectrum and the size

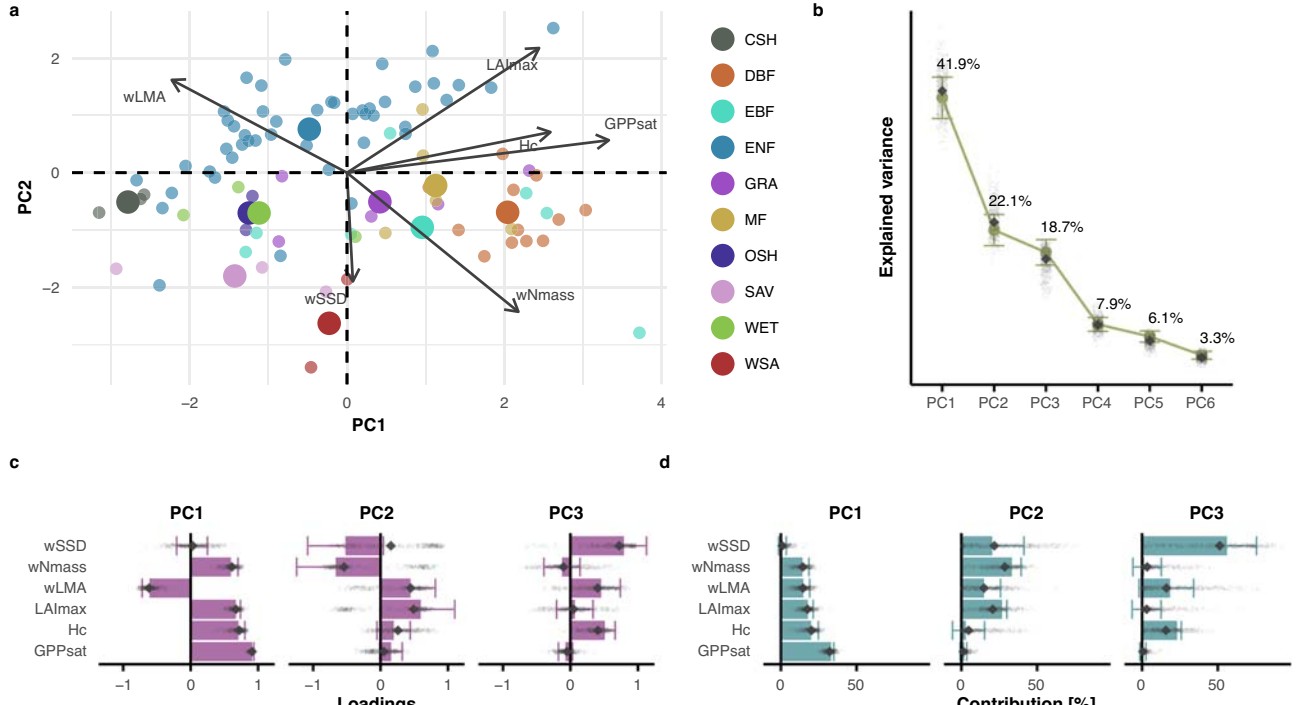

**Fig. 2 | Principal component analysis on the global spectrum of plant form and function at the ecosystem scale (89 sites). a** Biplot resulting from PCA; point colors represent plant functional types following the IGBP classification: CSH (Closed Shrubland), DBF (Deciduous Broadleaf Forest), EBF (Evergreen Broadleaf Forest), ENF (Evergreen Needleleaf Forest), GRA (Grassland), MF (Mixed Forest), OSH (Open Shrubland), SAV (Savannah), WET (Wetland), WSA (Woody Savannah). Bigger points represent the centroid of the distribution for each vegetation type. **b** Explained variance for the retained principal components (PCs). **c** Barplot for the loadings, and **d** contributions for each variable on the retained PCs. The full circles

in **b** and the bars in **c** and **d** show the pertinent estimate based on the full dataset. In **b–d**, the error bars are centered on the estimates and represent the standard error estimated with the bootstrap procedure ($n = 499$ bootstrap iterations); the small gray diamonds show the estimates of each bootstrap iteration, and the big gray diamonds represent the median of all bootstrap iteration. Variable acronyms: gross primary productivity at light saturation (GPPsat), canopy height (Hc), maximum leaf area index (LAImax), community-weighted mean leaf mass per area (wLMA), community-weighted mean nitrogen per leaf mass (wNmass), community-weighted mean stem specific density (wSSD).

dimension of vegetation combined likely explain the photosynthetic performance of the ecosystems. This would also explain why the eigenvectors of photosynthetic capacity and ecosystem respiration in Fig. 1 are not aligned with the eigenvectors of nitrogen content, leaf longevity, and leaf mass per area, characteristic of the leaf economics spectrum. The fact that size and structural elements such as leaf area index and canopy height also contribute to PC1 highlights how ecosystem processes are affected by vegetation biomass. The leaf economics spectrum represented by wNmass and wLMA is a trade-off between photosynthetic performance and structural persistence. Accordingly, the contributions of wNmass and wLMA are equally distributed between PC1 and PC2. The third PC was dominated by stem-specific density, with a $56.1 \pm 24.3\%$ contribution and a strong positive loading ($0.79 \pm 0.33$, Fig. 2, Supplementary Data 2). Canopy height and leaf mass per area also had important positive effects on PC3, underlying the importance of structural variables as important properties that emerge at the scale of ecosystems even beyond the plane of the global spectrum.

In the results based on the forest sites, the number of retained PCs was two, and only one when considering exclusively evergreen needleleaf forest sites. In these subcases, the plane between the performance-persistence trade, and the size axis, was less pronounced (Supplementary Fig. 4, Supplementary Data 2).

Considering that we could not include measures related to seed mass among the variables in our study and that we included the photosynthetic capacity to represent the ecosystem-level properties, we found a remarkable resemblance to the global spectrum study at the leaf scale[1]. However, we highlight one main difference: the effect of stem-specific density is partitioned between the second and especially the

third component. This difference could result from a bias in the dominant vegetation type. In particular, several dominant conifer species at some sites can have particularly low reported values of stem-specific density, and thus help shape a gradient from low-SSD grasslands and evergreen needleleaf forests, to high-SSD savannas, woody savannas, deciduous and evergreen broadleaf forests. However, wSSD is a weighted measure among all available species at a site, while canopy height is based on the maximum values of single individuals. Thus, the relationship between stem-specific density and plant height, characteristic of the size axis described by Díaz et al. might break down, with the community-weighted measure of stem-specific density having potentially a less clear ecological meaning than its plant-level counterpart.

The high number of retained axes (6) shows that multiple dimensions need to be considered when performing such analyses at the ecosystem scale. In particular, additional dimensions (beyond the second component) could hint at secondary effects of e.g., water transport within the soil–plant–atmosphere continuum, or water storage. In fact, canopy height and stem-specific density are indirectly linked to plant hydraulics, especially in trees. For instance, canopy height relates to the water potentials in the plant and is inversely proportional to the transpiration rate in Darcy's law[26]. At the same time, canopy height and stem-specific density in trees are constrained by hydraulic limitations such as cavitation risk[27]. The additional relationships uncovered by the third component could characterize how water is transported through the plant vessels and stored in wood tissues. In the following section related to the least-cost hypothesis, we show that the dimension related to water is indeed important. Other hidden mechanisms that are not apparent with this set of variables, such as soil chemical and physical characteristics, likely play an

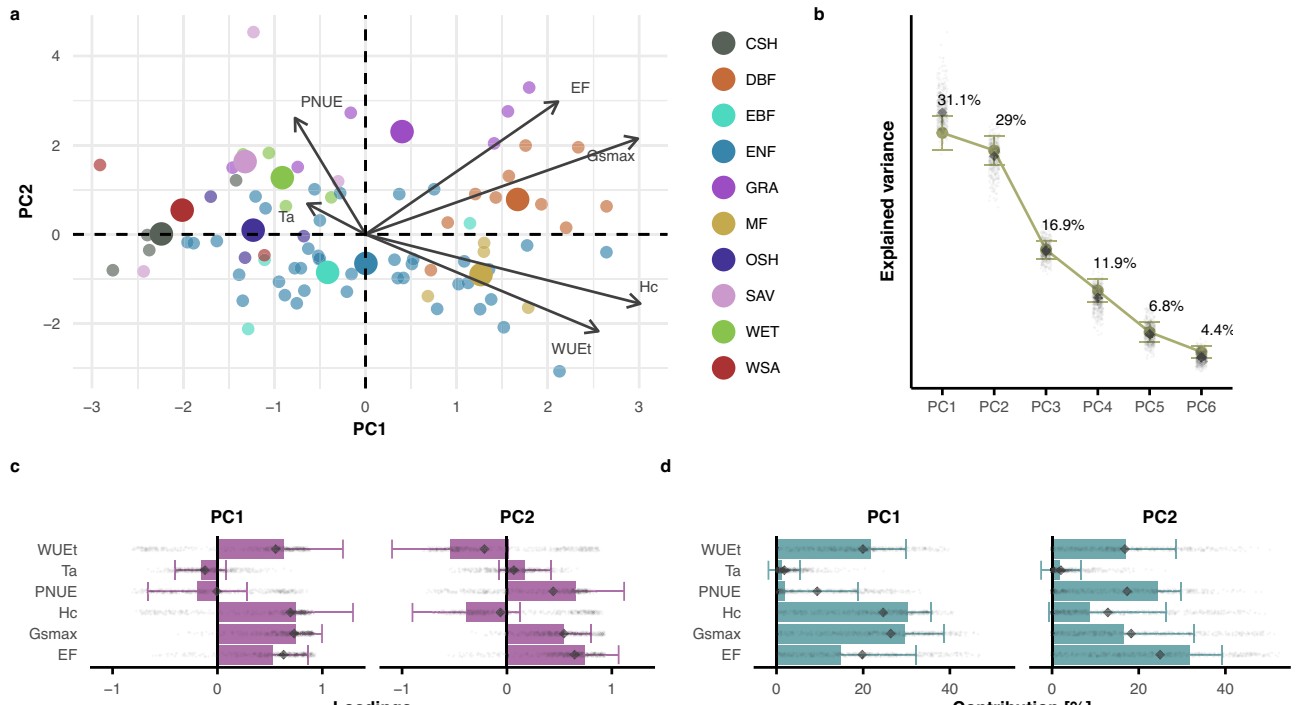

**Fig. 3 | Principal component analysis on analogous variables of the least-cost hypothesis (82 sites). a** Biplot resulting from PCA; point colors represent plant functional types following the IGBP classification: CSH (Closed Shrubland), DBF (Deciduous Broadleaf Forest), EBF (Evergreen Broadleaf Forest), ENF (Evergreen Needleleaf Forest), GRA (Grassland), MF (Mixed Forest), OSH (Open Shrubland), SAV (Savannah), WET (Wetland), WSA (Woody Savannah). Bigger points represent the centroid of the distribution for each habitat type. **b** Explained variance for the retained principal components (PCs). **c** Barplot for the loadings, and **d** contributions for each variable on the retained PCs. The full circles in **b** and the bars in **c** and **d** show the pertinent estimate based on the full dataset. In **b**–**d**, the error bars are centered on the estimates and represent the standard error estimated with the bootstrap procedure ($n = 499$ bootstrap iterations); the small gray diamonds show the estimates of each bootstrap iteration, and the big gray diamonds represent the median of all bootstrap iteration. Variable acronyms: evaporative fraction (EF), maximum surface conductance (Gsmax), canopy height (Hc), maximum leaf area index (LAImax), ecosystem-scale photosynthetic nitrogen use-efficiency (PNUE), air temperature (Ta), water use-efficiency based on transpiration (WUEt).

important role in defining PCs beyond the first two dimensions. Still, we would need consistent measurements across the network to resolve such limitations[28,29].

Our results are partly in line with recent literature describing three main components related to productivity, water, and carbon use[30]. Compared to Migliavacca et al., we include additional structural and chemical variables, and we find similar components for productivity and water properties in our analog of the global spectrum of plant form and function. Additionally, we explain a higher proportion of variance over the first three components and highlight the importance of the structure and size of vegetation within the ecosystem as a secondary but crucial component.

The exclusion of non-forest sites leads only to minor changes to our results, in line with a recent study that expands on trait coordination at the leaf scale and compares woody and non-woody species[31].

While previous studies already highlighted the striking similarity of community-level relationships compared to the plant-level trade-offs of Díaz's global spectrum[12], we find that the same is true for whole-ecosystem relationships between community-weighted averages of plant traits and ecosystem functions inferred using surface–atmosphere measurements. We conclude that the same eco-evolutionary constraints affecting individual plant fitness and community assemblages also apply to whole ecosystems.

**Least-cost hypothesis at the ecosystem scale**
For the analyses of the least-cost hypothesis[7,11] at the ecosystem scale, we focused primarily on the expected trade-off between the costs in the acquisition, retention, and use-efficiencies of nitrogen and water. Therefore, we considered variables directly or indirectly related to the

costs of nitrogen (photosynthetic nitrogen use-efficiency—PNUE), and water (directly related: water use-efficiency—WUEt, maximum stomatal conductance—Gsmax, evaporative fraction—EF; indirectly: air temperature—Ta, canopy height—Hc, Supplementary Table 1). Results of the PCA showed strong positive loadings on the first component for almost all variables (Fig. 3). In fact, PC1 represented the dimension of the maximum rates of processes, i.e., the performance dimension that we identify with properties related to productivity and metabolic rates (e.g., photosynthesis, respiration, water and gas exchange). This was consistent when including GPPsat or different metrics related to productivity and photosynthetic nitrogen use efficiency (e.g., Supplementary Fig. 5, Supplementary Fig. 6). The fact that variables such as canopy height, maximum surface conductance, evaporative fraction, and water use-efficiency have positive loadings on PC1 indicates that this component reflects the gradient between low stature vegetation with limited available resources and low maximum surface conductance (e.g., water-limited, or low temperature) to high stature vegetation with high water availability. The effect of PNUE is inversely related to the other variables, in line with the least-cost hypothesis, but hardly relevant on PC1 (Fig. 3). The dimension of maximum rates consistently emerges at the ecosystem scale, regardless of the set of chosen variables (Supplementary Fig. 5, Supplementary Fig. 6). The second PC explained $29 \pm 1.8\%$ of the variance (Fig. 3b) and uncovered the trade-offs expected by the least-cost hypothesis: a negative relationship between the loadings of water use-efficiency and canopy height on one side (WUEt: $-0.54 \pm 0.55$, and Hc: $-0.39 \pm 0.51$), and evaporative fraction, photosynthetic nitrogen use-efficiency, and surface conductance on the other side (EF: $0.74 \pm 0.32$, PNUE: $0.65 \pm 0.47$, and Gsmax: $0.54 \pm 0.26$, Fig. 3c, d, Supplementary Data 3).

Tests with alternative metrics of water use-efficiency and photosynthetic nitrogen use-efficiency confirmed the negative relationship between these two ecosystem properties on the second PC of variability (Supplementary Fig. 5, Supplementary Fig. 6, and Supplementary Data 3). However, the difference in $R^2$ for PC1 and PC2 increased substantially when using alternative formulations of the two ecosystem properties, or different subsets of the data, suggesting that the first component related to performance dominates the functional space of ecosystem properties (Supplementary Figs. 5–8). Differences between the full estimates and the median of the bootstrap iterations could be related to dataset biases in terms of a disproportionate number of forest sites, and in particular evergreen needleleaf forests, so we again repeated the analysis on these subsets of sites. However, our analysis on all forest sites or evergreen needleleaf forests produced similar results as for the overall case with all sites. The directionality of the relationships between variables was similar to the overall results, albeit less pronounced (Supplementary Fig. 9, Supplementary Data 3).

The first component is consistent with earlier leaf-level studies showing a positive relationship between the maximum rates of processes (e.g., surface conductance, or net photosynthesis), structural variables (e.g., leaf area index, or specific leaf area, the inverse of leaf mass per area), and foliar chemistry (leaf nitrogen)[4,32]. At the ecosystem scale, PNUE and Gsmax feature a positive directionality in the first component, in line with the notion that these two variables positively affect productivity in the context of the leaf economics spectrum[33]. Our results also show the negative relationship between PNUE and WUEt on the second dimension of the PCA, as expected from leaf-level field studies and theory[2,7,11,34–36]. Additionally, other expected trade-offs are present on this component, such as the negative relationship between surface conductance and water use-efficiency[37], or a negative relationship between WUEt and evaporative fraction, which is low at more arid sites and higher at wet sites. This is in line with the expected increase in the efficiency of plants in using water along aridity gradients, as shown with leaf-level measurements of leaf-internal to ambient $CO_2$ ratio as a proxy of intrinsic water use-efficiency[38,39]. In sum, the second component in our third and final analysis unravels the axis of the least-cost hypothesis. The coordination between the variables of the least-cost hypothesis covers a range of sites from wet conditions with high efficiency of photosynthetic nitrogen use, but low water use-efficiency, to arid conditions with high efficiency of water use, but low photosynthetic nitrogen use-efficiency. Measures of leaf-internal and ambient $CO_2$ mole fraction, or stable carbon isotope signatures measured at the sites would help to strengthen our claims related to the least-cost hypothesis, but these measurements were unavailable for the large majority of sites.

Overall, we argue that the maximum rates related to productivity dominate ecosystem functioning, while the least-cost hypothesis only emerges as a secondary, yet still important, trade-off. In this context, the dimension of productivity could be described as a scale-emergent property at the ecosystem level. Furthermore, our definition of some ecosystem-level metrics included aspects that are not required or even appropriate at the leaf or plant scale. For instance, the distinction between transpiration and evaporation needs to be considered when computing the water use-efficiency from eddy covariance fluxes at the ecosystem scale. Consequently, the leaf area index needs to be included in the calculation of photosynthetic nitrogen use-efficiency. These effects are scale-emergent properties, meaning that evaporation or leaf area index are not prominent properties for leaf-level processes, but they are key at the ecosystem scale. At this scale, scale-emergent properties weaken the relationship between the variables connected to the least-cost hypothesis, when not properly accounted for.

The relationships underlying the least-cost hypothesis might therefore not always be conserved at the ecosystem scale, which can be explained by multiple reasons. First, some of the previous studies on the least-cost hypothesis generally focused on limited geographical ranges[7,11,38]. Our dataset displays much stronger variation in plant resource use patterns along axes of nutrient availability and disturbance. However, global-scale evidence for the least-cost hypothesis also exists based on modeling[40,41], or global measurements of leaf-level carbon isotopes[42]. In our analysis, this optimality principle might be more elusive at the global scale, because we simultaneously characterize other trade-offs on the main axis—the component of maximum rates—which dominates the gradient in average ecosystem functional properties[30,32]. The least-cost hypothesis is only observed when this effect is removed, which is not evident in leaf-level studies, and which could be considered a scale-emergent behavior at the ecosystem scale. Second, ecosystems are a mix of different individuals and species, with different phenologies and different physiological statuses due to biotic and abiotic effects. This mix could limit the strength of the signal of leaf-level coordination theories at the ecosystem scale since optimization for one individual might not coincide with an averaged optimization for the whole ecosystem. For instance, Medlyn et al.[43] showed that it is difficult to reconcile leaf-level and ecosystem-scale estimates of water use-efficiency. Regardless of our different computations of WUEt based on transpiration, this suggests that a simple averaging or sum of the ecosystem components does not guarantee capturing the whole ecosystem response. Third, intraspecific variability might confound the ecosystem response. We did not explicitly account for intraspecific variation and aggregated our metrics to a unique average (or maximum) value at each site. For instance, Dong et al.[38] demonstrated that most variation in the ratio of intracellular to atmospheric $CO_2$ concentration is expressed within species. In general, plant strategies are species-specific, and quite plastic to changes in environmental drivers. We argue that a combination of species with different life histories at globally distributed sites may not necessarily average to a single common trade-off of water and nitrogen cost minimization.

The potential confounding factors outlined above apply to all parts of our analysis. However, these confounding factors might only be worth considering when the signal of the relationships between variables is already overshadowed by a more dominant component. For instance, the trade-offs underlying the least-cost hypothesis are eclipsed by the dimension of maximum rates of ecosystem processes.

## Caveats and implications

Leaf-level coordination principles propagate to the ecosystem scale. In particular, we show strong evidence supporting the hypothesis that the leaf economics spectrum is conserved at the ecosystem level. The global spectrum of plant form and function and the least-cost hypothesis are also evident for whole ecosystems, despite embodying secondary mechanisms at the ecosystem scale.

However, by upscaling the leaf-level coordination principles to the ecosystem scale, we also observe higher complexity, as suggested by an increase in significant PCs compared to those identified by the original theories at the leaf scale[1,8]. Certain aspects of trait coordination are conserved at the ecosystem scale (e.g., the relationship between photosynthetic performance and leaf persistence of the leaf economics spectrum[8]). Conversely, other trade-offs might be more elusive due to a set of potential issues underlying the data, due to scale-emergent properties (e.g., structure or evaporation), or due to properties intrinsic to ecosystem-level processes (e.g., optimization of nitrogen use and water use is a secondary dimension). Therefore, accounting for potential confounding factors such as canopy structure, leaf area index, or processes such as evaporation is important for an accurate representation of ecosystem-level processes and relationships in ecological theory and dynamic global vegetation models (DGVMs). DGVMs usually rely on constant vegetation parameters (e.g., mean traits) to simulate changes in carbon stocks (e.g., LAI) and ecosystem processes and fluxes. The DGVMs parameters are constant per plant functional type: for example, LMA or N content in leaves are

parameterized as the mean values for large plant functional type classes such as deciduous, or evergreen forests. This parameterization typically neglects the variation in traits and the coordination between traits and functions observed in nature. Instead, ecosystem functions (e.g., GPPsat, RECOmax) are simulated as a response to foliage density (related to LAImax). This current paradigm is not flexible enough to represent the variability and coordination between traits and functions and therefore can lead to biases in modeling[30]. For instance, for the leaf economics spectrum, we can use a linear mixed model to test the relationship between GPPsat or RECOmax, the foliar traits (wLMA, wNmass, wLL), and the covariation between the variables once accounted for vegetation class and leaf area index as random effects. With this test, we showed that some of the fixed effects resembling the trade-offs in the leaf economics spectrum at the ecosystem scale are important even when accounting for leaf area index and plant functional type, and should therefore not be overlooked (Supplementary Table 2). Recent studies focusing on DGVMs development are focusing on further including coordination principles, with explicit covariation of trait and functional parameters within vegetation cover classes[22]. In this sense, our analysis can help to indicate which traits and functions can be helpful in supporting the current developments.

We acknowledge some potential shortcomings in our study. First, a mismatch between site-level conditions and plant traits from secondary data sources is possible, since plant trait values from databases do not necessarily represent adaptations to the local site conditions (e.g., LL, LMA, SSD). However, some encouraging results indicate that this may not be a major issue. In the case of leaf nitrogen, a recent study showed that it is possible to use the TRY database and maintain robust relationships with ecosystem-scale GPPsat[24]. Moreover, for European forests, it is possible to use traits from the TRY database and obtain very similar community-weighted means compared to the in-situ data[44]. Second, the different lengths of flux measurements available at the site level impact the calculation of the ecosystem functional properties, particularly for sites with extreme weather conditions and few years of data. We accounted for this shortcoming by selecting the maximum (or potential) value of ecosystem functional properties (e.g., GPPsat, Gsmax) within the measurement period. Within the relatively short study periods of most eddy covariance sites, this should minimize the mismatches in species representativeness of plant traits and the effects of meteorological variability on the fluxes.

Our results demonstrate that fundamental leaf- and plant-level coordination principles propagate to the ecosystem scale. The same drivers forcing plant trait expression also shape the functioning of whole ecosystems. However, scale-emergent properties should be carefully considered when looking at ecosystem-level phenomena, because they can partly mask the scaling of leaf-level coordination principles. Additionally, even though coordination principles are important for whole ecosystems, they might be masked by more dominant relationships, such as the dimension of the maximum rates of processes. Future studies on ecosystem-level optimality should focus on increasing the number of sites, prioritizing underrepresented bioclimatic regions (e.g., tropics), and on the refinement of vegetation properties and other important stand characteristics, including soil properties[31]. In this context, our original hypothesis that the known leaf-level coordination of functional traits is conserved at the ecosystem level should be further investigated with additional case studies. Furthermore, dynamic global vegetation models should be tested with and without optimality included[9,22,41,45]. Considering the increasing effort to include optimality principles in the land surface scheme of Earth system models, we suggest using our approach and results as a benchmark for model runs. The validation of established optimality principles at different scales would support more accurate implementation of the notions learned from leaf-level theories in models across scales.

## Methods

### Eddy covariance FLUXNET sites

We used data from the global network of eddy covariance flux tower stations (FLUXNET), integrating the LaThuile dataset[46] with the FLUXNET2015 dataset[47]. In case of overlap of sites in the two datasets, the FLUXNET2015 dataset was used. We excluded cropland sites in order to avoid the influence of intense management practices (irrigation, plowing, fertilization, etc.). The dataset used for the analysis included sites with more than 3 years of data and the availability of ancillary data described below. The selected 98 sites cover different biomes and climate zones: from tropical, Mediterranean, temperate, and boreal to arctic sites, including major forest types, grasslands, savannas, shrublands, and wetlands (Supplementary Data 4, Supplementary Table 3).

### Plant traits and vegetation properties

For each FLUXNET site, we collected a set of plant traits for constituent species or site means (leaf longevity, leaf mass per area, nitrogen per leaf area, nitrogen per leaf mass, and stem-specific density), and site-level vegetation characteristics (canopy height, maximum leaf area index) from the FLUXNET or Ameriflux ancillary data, or, if not reported, directly from site principal investigators. Where site measurements were unavailable, we included information from the TRY database[48] (a full list of plant traits data sources can be found in Supplementary Data 5) or data from the literature for the specific sites[30,49,50]. We obtained site constituent species and species abundances at the sites (percentage of area covered by each species) from the literature[49–53], and by consulting site principal investigators. We assumed homogeneous distribution for species with missing abundance information, following the approach described in previous studies[51,52]. We excluded sites where the total sum of known species abundances was below 50% of the total site area.

For each site, we computed the community mean weighted by species fractional cover[54] for the following plant traits: leaf longevity (wLL, months), leaf dry mass per area (wLMA, mg mm$^{-2}$), stem specific density (wSSD, g cm$^{-3}$, with SSD defined as stem dry mass per stem fresh volume), and nitrogen per leaf mass (wNmass, %). For some sites, site principal investigators provided site-level estimates of plant traits upscaled with similar methodologies, which were prioritized over TRY-derived estimates. Regarding leaf longevity, we could not account for different leaf age groups because of a lack of data.

We calculated weighted nitrogen per leaf area as the product of wNmass and wLMA (wNarea, g N m leaf$^{-2}$). We collected canopy height (Hc, m) and maximum leaf area index (LAImax, m$^2$ m$^{-2}$) from FLUXNET or Ameriflux ancillary data products, site principal investigators, and the literature[30,50].

### Eddy covariance fluxes and ecosystem functional properties

We calculated ecosystem functional properties from carbon, water, energy fluxes, and meteorological data measured or estimated at half-hourly/hourly time steps at the selected FLUXNET sites. Supplementary Table 1 provides a comparison between leaf- and plant-level traits and the analogous ecosystem functional properties and vegetation properties used in this study, while Supplementary Table 4 lists all the variables used in the computations of ecosystem functional properties. We used gross primary production (GPP) and ecosystem respiration (RECO) estimated from measured net ecosystem exchange (NEE) using the night-time partitioning method[55]. The methodology for the calculation of each ecosystem's functional properties used in this study is described below.

We retained data with good quality (quality check 0−measured data, and 1−good quality gap-filled data), and, additionally, we retained the data measured during the active growing season, determined as the period when daily GPP is above the 30% of the difference between maximum and minimum daily GPP. For each site, we

aggregated the filtered half-hourly/hourly data to mean yearly values for air temperature, vapor pressure deficit, and incoming shortwave radiation (SWin, W m$^{-2}$), and mean yearly cumulative values for precipitation. Transpiration flux estimates were calculated following the methodology in Nelson et al.[56]. We collected elevation information for each site from the FLUXNET or Ameriflux Biological, Ancillary, Disturbance, and Metadata (hereafter: ancillary data), FLUXNET websites, and the OzFlux website for one Australian site.

**Photosynthetic capacity (GPPsat).** We filtered half-hourly/hourly flux data based on SWin to exclude night-time values (SWin > 100 W m$^{-2}$). We fitted GPP and SWin to a hyperbolic light response curve with a moving window of 5 days, with the values assigned as the center of the moving window[30]. For each moving window, we extracted the photosynthetic capacity at light saturation (GPPsat, μmol CO$_2$ m$^{-2}$ s$^{-1}$)[24,53,57,58] as the value of the fitted functions at a saturating photosynthetic photon flux density of 2000 μmol m$^{-2}$ s$^{-1}$. The photosynthetic photon flux density was derived as SWin * 2.11[59]. We excluded GPPsat estimates above a threshold of 60 μmol CO$_2$ m$^{-2}$ s$^{-1}$ to omit unrealistic values of GPPsat according to the distribution of GPPsat. For each year and growing season, we extracted the 95th percentile from the GPPsat estimates. The 95th percentile was chosen because the calculation of GPPsat based on a fitted model had less noise than other ecosystem functional properties. For each site, we used GPPsat as the average over the yearly 95th GPPsat percentiles.

**Maximum ecosystem respiration (RECOmax).** We filtered half-hourly/hourly flux data based on SWin to exclude day-time values (SWin < 50 W m$^{-2}$). For each site, we considered the 90th percentile of night-time net ecosystem exchange as a measure of maximum ecosystem respiration (RECOmax, μmol CO$_2$ m$^{-2}$ s$^{-1}$).

**Evaporative fraction (EF).** From the half-hourly/hourly flux data, we removed periods with precipitation events and the following 48 h (where available, P < 0.1 mm). We also excluded night-time values (SWin > 200 W m$^{-2}$). We included a filter based on friction velocity (u* > 0.20 m s$^{-1}$)[60] to minimize the use of records potentially affected by flux underestimation. We computed EF (unitless) as the ratio of latent heat flux (W m$^{-2}$) to the available energy flux which was approximated by the sum of latent and sensible heat fluxes (W m$^{-2}$)[61]. For each site, we used the median of EF over the available measurement period.

**Maximum surface conductance (Gsmax).** We retained half-hourly/hourly flux data with the same filters described for the calculation of EF, and we additionally excluded noisy measurements with negative values of vapor pressure deficit. We computed the aerodynamic conductance for heat transfer (Ga, m s$^{-1}$), and calculated the surface conductance (Gs, m s$^{-1}$) by inverting the Penman–Monteith equation, using the *bigleaf* R package[60] and following the methodology in Migliavacca et al.[30]. For each site, we computed the maximum surface conductance (Gsmax, m s$^{-1}$) as the 90th percentile of Gs values over the available measurement period.

**Photosynthetic nitrogen use-efficiency (PNUE).** We computed photosynthetic nitrogen use-efficiency (μmol CO$_2$ g N$^{-1}$ s$^{-1}$) as PNUE = GPPsat/(wNarea * LAImax), based on the formulation from the literature[2,7]. We accounted for the scaling to the ecosystem level by including LAImax.

**Water-use efficiency (WUEt).** We filtered half-hourly/hourly flux data based on potential incoming shortwave radiation to exclude night-time values (SWin_pot > 200 W m$^{-2}$). We then aggregated the data to daily values. We filtered the daily-aggregated flux data based on the fraction of good quality data (fraction > 0.8 of NEE quality check

0–measured data, and 1–good quality gap-filled data[47]. We excluded entries where the daily ratio of GPP to T exceeded the site mean by three times the standard deviation, where T refers to the transpiration estimates provided by the TEA algorithm[56]. We calculated the water-use efficiency based on transpiration in order to avoid confounding effects from evaporation (WUEt, μmol CO$_2$ mmol H$_2$O$^{-1}$). For each site, we computed WUEt as the ratio of cumulative GPP to cumulative T over the period of available filtered data.

## Statistical analysis

For each of the three hypotheses examined, we conducted PCA on selected variables, in order to avoid clustering of a priori known strong correlations, reduce the dimensionality of the datasets, and thereby increase the interpretability of the data[62]. In the PCA results, the sign and direction of the eigenvectors denote relationships and trade-offs between the variables (arrows in panels a of Figs. 1–3). Eigenvectors that are orthogonal to one another suggest trade-offs between the corresponding variables, while parallel eigenvectors indicate correlations. The same concept applies to the loadings, which represent the projections of the eigenvectors on each PC: loadings with a different sign highlight potential trade-offs between variables, and equal signs indicate potential correlations (panels c in Figs. 1–3). A highly explained variance assures that the selected variables are appropriately capturing the variance in the data for each PC and as a whole (panels b of Figs. 1–3). Finally, the contributions describe how variables define each PC (panels d of Figs. 1–3). We used the *PCA* function in the *FactoMineR* R package[63]. For standardization, we applied z-transformation to each variable. For each section of the analysis, we tested the number of significant PCs to be retained following Dray's method[25], using the *ade4* R package[64,65], in order to minimize redundancy as well as loss of information.

For each section of the analysis, we obtained subsets of the dataset via substitution bootstrapping using the *bootstrap* function in the *modelr* R Package[66] (499 replicates). We repeated the PCA on the bootstrapped datasets and then computed the standard deviation from the bootstrapped outputs to obtain the bootstrap standard error for the explained variance, contributions, and loadings. For all analyses, we repeated the test for (1) all available sites in our dataset, (2) forest sites only, and (3) evergreen needleleaf forest sites only.

Based on the output models of multi-model inference[67] via the *dredge* function in the *MuMIn*[68] R package, we conducted relative importance analysis using the *calc.relimp* function of the *relaimpo*[69] R package. This was done to evaluate the importance of predictors of GPPsat.

Finally, we used the *lmer* function in the *lme4*[70] R package to fit a linear mixed model to predict GPPsat and RECOmax based on wLMA, wNmass, and wLL as fixed effects, and adding a random slope on the predictor, i.e., the (random) effect of LAImax, for each vegetation class (IGBP). This was fed to the model function as: y -wNmass + wLMA + wLL + (LAImax | IGBP), on a sample size of 87 sites.

## Inclusion and ethics

All relevant data contributors have been invited to participate in the manuscript preparation, and given co-authorship where appropriate, or have otherwise been appropriately acknowledged.

## Reporting summary

Further information on research design is available in the Nature Portfolio Reporting Summary linked to this article.

## Data availability

The processed eddy covariance data–the LaThuile dataset (https://fluxnet.fluxdata.org/data/la-thuile-dataset/) and the FLUXNET2015 dataset (https://fluxnet.fluxdata.org/data/fluxnet2015-dataset/)–are

available on the FLUXNET website. Biological, Ancillary, Disturbance, and Metadata for the sites are available in the respective databases (https://fluxnet.org/data/fluxnet2015-dataset/, https://fluxnet.org/data/la-thuile-dataset/, https://ameriflux.lbl.gov/data/badm/, https://ameriflux.lbl.gov/sites/site-search/, and https://www.ozflux.org.au/monitoringsites/calperum/calperum_dem.html) and in the cited literature. The plant traits measurements data are available on the TRY database (https://www.try-db.org/TryWeb/Home.php) either publicly or under restricted access due to embargo and can be obtained via request on the TRY platform. The data necessary to interpret, verify, and extend the research in this article are available in the Zenodo database under the accession code https://doi.org/10.5281/zenodo.7984734.

## Code availability

All the analyses were conducted with R 4.1.0 for Windows (64-bit). The R package used for the calculation of the ecosystem functional properties is already described in the literature and freely available on CRAN: bigleaf v0.8.2 (https://cran.r-project.org/web/packages/bigleaf/). The R code used for the statistical analyses uses packages available on CRAN: FactoMineR v2.6 (https://cran.r-project.org/web/packages/FactoMineR/), ade4 v1.7-20 (https://cran.r-project.org/web/packages/ade4/), modelr v0.1.9 (https://cran.r-project.org/web/packages/modelr/), MuMIn v1.43.17 (https://cran.r-project.org/web/packages/MuMIn/), relaimpo v2.2-6 (https://cran.r-project.org/web/packages/relaimpo/), and lme4 v1.1-31 (https://cran.r-project.org/web/packages/lme4/). The TEA algorithm v1.1 is available at https://doi.org/10.5281/zenodo.3921923. The R codes used for this analysis are available on Zenodo at https://doi.org/10.5281/zenodo.7984734.

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

## Acknowledgements

This work used eddy covariance data acquired and shared by the FLUXNET community, including these networks: AmeriFlux, AfriFlux, AsiaFlux, CarboAfrica, CarboEuropeIP, CarboItaly, CarboMont, China-Flux, Fluxnet-Canada, GreenGrass, ICOS, KoFlux, LBA, NECC, OzFlux-TERN, Swiss FluxNet, TCOS-Siberia, and USCCC. The FLUXNET eddy covariance data processing and harmonization was carried out by the ICOS Ecosystem Thematic Center, AmeriFlux Management Project, and Fluxdata project of FLUXNET, with the support of CDIAC, and the OzFlux, ChinaFlux, and AsiaFlux offices. We thank Weber U., Holst J., Meyer W., Hughes H., Nave L., Kosugi Y., Stuart-Haëntjens E., Arndt S., Battles J., Desai A., Moore T., Vogel C., Munger W. J., and York R. for contributing data used in this study. U. Gomarasca and M. Migliavacca thank the International Max Planck Research School (IMPRS). M. Reichstein and G. Duveiller acknowledge funding by the European Research Council (ERC) Synergy Grant "Understanding and Modeling the Earth System with Machine Learning (USMILE)" under the Horizon 2020 research and innovation program (Grant agreement No. 855187). This work was also supported by the Swiss National Science Foundation (40FA40_154245; 20FI21_148992; 20FI20_173691; 20FI20_198227) to N. Buchmann, the National Research Foundation of Korea (NRF) grant funded by the Korea government (MSIT) (2022R1A2C1003504) to C. Byun, the German Research Foundation DFG (INST 186/1118-1 FUGG) and the Ministry of Lower-Saxony for Science and Culture (DigitalForst: Niedersächsisches Vorab, ZN 3679) to A. Knohl, the Estonian Research Council team grant

PRG537 to Ü. Niinemets, the Organismo Autónomo de Parques Nacionales (project 2822/2021) to O. Perez-Priego, the Spanish Government grant PID2019-110521GB-I00 to J. Peñuelas; the U.S. National Science Foundation, Biological Integration Institutes grant NSF-DBI-2021898 to P. B. Reich, the CzeCOS program (grant number LM2018123) and SustES—Adaptation strategies for sustainable ecosystem services and food security under adverse environmental conditions (CZ.02.1.01/0.0/0.0/16019/0000797) to L. Šigut, and European Space Agency Living Planet Fellowship 'Vad3e mecum' to S. Walther.

## Author contributions

U.G. and M.Mi. conceptualized the study, collected and processed the data, wrote the code and performed the analysis, and wrote and revised the draft. J.K., Ü.N., C.W., A.Ce., M.Ba., R.N., G.D., M.Ma., J.P., O.P.P., P.B.R., I.J.W., and M.R. substantially contributed in conceptualizing the study, designing the analysis, interpreting the results, or supervising. J.K., M.Ba., J.P., J.A.N., Ü.N., A.T.R.C., M.A.A., M.Be., T.A.B., H.H.B., S.F.B., N.B., A.Ca., B.C., A.Co., A.C.d.S., S.F., A.I., A.K., B.K., J.M.L., C.H.L., V.M., L.M., A.M., Y.O., P.P., T.L.P., L.S., P.M.v.B., and G.W. helped to collect or process the data. J.A.N., D.M., and S.W. performed or helped with part of the analyses. All authors contributed to the revision of the final paper.

## Funding

## Competing interests

The authors declare no competing interests.

## Additional information

[1]Max Planck Institute for Biogeochemistry, Hans-Knöll-Str. 10, 07745 Jena, Germany. [2]European Commission, Joint Research Centre, Ispra 21027 VA, Italy. [3]Chair of Plant and Crop Science, Estonian University of Life Sciences, Kreutzwaldi 1, 51006 Tartu, Estonia. [4]Institute of Biology, Leipzig University, Leipzig, Germany. [5]German Centre for Integrative Biodiversity Research (iDiv) Halle-Jena Leipzig, Leipzig, Germany. [6]Universität Innsbruck, Institut für Ökologie, Innsbruck, Austria. [7]Discipline of Botany, School of Natural Sciences Trinity College Dublin, Dublin, Ireland. [8]Dipartimento di Scienze - Università Roma TRE - V.le Marconi 446, 00146 Roma, Italy. [9]School of Earth, Environment & Society and McMaster Centre for Climate Change, McMaster University, Hamilton, ON, Canada. [10]Institute of Terrestrial Ecosystems, ETH Zurich, Zurich, Switzerland. [11]Faculty of Land and Food Systems, University of British Columbia, Vancouver, BC, Canada. [12]Department of Biology, University of Copenhagen, Universitetsparken 15, 2100 Copenhagen Ø, Denmark. [13]Institute of Ecology and Evolution - Friedrich Schiller University Jena, Philosophenweg 16, 07743 Jena, Germany. [14]Department of Environmental Systems Science, ETH Zurich, Zurich, Switzerland. [15]Department of Biological Sciences, Andong National University, Andong 36729, Republic of Korea. [16]Fundación Centro de Estudios Ambientales del Mediterráneo (CEAM), Paterna, Spain. [17]National Research Council of Italy (CNR), Institute for Sustainable Plant Protection (IPSP), Metaponto 75012, Italy. [18]Santa Catarina State University, Agroveterinary Center, Forestry Department, Av Luiz de Camões, 2090, Conta Dinheiro, 88.520-000, Lages, SC, Brazil. [19]National Research Council of Italy (CNR), Institute for Agriculture and Forestry Systems in the Mediterranean (ISAFOM), Naples 80055, Italy. [20]Technical University of Denmark (DTU), Environmental Engineering and Resource Management, Bygningstorvet 115, 2800 Kgs. Lyngby, Denmark. [21]Bioclimatology, University of Göttingen, Büsgenweg 2, 37077 Göttingen, Germany. [22]Andorra Research + Innovation; Avinguda Rocafort 21-23, Edifici Molí, 3r pis, AD600 Sant Julià de Lòria, Andorra. [23]CEFE, Univ Montpellier, CNRS, EPHE, IRD, Montpellier, France. [24]Environmenal Research Institute, University of Waikato, Private Bag, 3105 Hamilton, New Zealand. [25]Remote Sensing Centre for Earth System Research, Leipzig University, 04103 Leipzig, Germany. [26]Department of Biology, Vrije Universiteit Brussel, Pleinlaan 2, 1050 Brussel, Belgium. [27]Faculty of Science and Technology, Free University of Bolzano, Piazza Università 5, 39100 Bolzano, Italy. [28]Research Center for Advanced Science and Technology, the University of Tokyo, 4-6-1 Komaba, Meguro, Tokyo 153-8904, Japan. [29]Graduate School of Agriculture, Kyoto University, Oiwake, Kitashirakawa, Kyoto 606-8502, Japan. [30]CSIC, Global Ecology Unit CREAF-CSIC-UAB, Bellaterra, Barcelona 08193 Catalonia, Spain. [31]CREAF, Cerdanyola del Vallès, Barcelona 08193 Catalonia, Spain. [32]Department of Forestry Engineering, University of Córdoba, Edif. Leonardo da Vinci, Campus de Rabanales s/n, 14071 Córdoba, Spain. [33]Ecology and Conservation Biology, Institute of Plant Sciences - Faculty of Biology and Preclinical Medicine - University of Regensburg, Universitaetsstrasse 31, D-93053 Regensburg, Germany. [34]The Department of Earth and Environmental Systems, The University of the South, Sewanee, TN, USA. [35]Department of Forest Resources, University of Minnesota, St. Paul, MN 55108, USA. [36]Institute for Global Change Biology, and School for Environment and Sustainability, University of Michigan, Ann Arbor, MI 48109, USA. [37]Hawkesbury Institute for the Environment, Western Sydney University, Penrith, NSW 2753, Australia. [38]Department of Matter and Energy Fluxes, Global Change Research Institute of the Czech Academy of Sciences, Bělidla 986/4a, 603 00 Brno, Czech Republic. [39]Institute of Environmental Sciences, Leiden University, Einsteinweg 2, 2333 CC Leiden, the Netherlands. [40]School of Natural Sciences, Macquarie University, Macquarie Park, NSW 2109, Australia. ✉e-mail: ugomar@bgc-jena.mpg.de

