## [Peer Review File · Nature Communications]

Leaf-level coordination principles propagate to the ecosystem
scaleREVIEWER COMMENTS

Reviewer #1 (Remarks to the Author):

In the manuscript “Leaf-level coordination principles propagate to the ecosystem scale”, Gomarasca et al. applied the principal component analysis (PCA) to multiple ecosystem variables derived from 98 flux sites. They tested whether the leaf economic spectrum, the global spectrum of form and function and the least cost hypothesis are valid at the ecosystem level, and concluded they are. The research question is well framed, and I can see it is natural follow-up to some earlier works from the group (Migliavacca et al. 2021). However, I found it was quite challenging to understand the meanings of each PCs and clearly grasp the implications of the study, in the current form of manuscript. I am afraid it still has a distance to go to reach the quality of nature communications.

1. While PCA is a useful method to identify the main type of factors that influence ecosystem functions, it is not necessarily clear to readers why higher explanatory power of first few PCs means there is traits coordination or tradeoff. Perhaps there is a need to clarify that in methods or introduction.
2. L105-106. I am not quite sure this is the only definition of the least cost theory, as some might have formulated the least cost hypothesis in a different way (e.g., Prentice 2014 Ecology Letters). As for water resource, I think the cost is not only about its “acquisition”, would also relate to the cost to prevent water from losing (especially in terms of water use efficiency).
3. L111-121. Do not quite follow here. I thought the paragraph is meant to introduce the concept of scale-emergent properties and then lead to few hypotheses on the existence of ecosystem level coordination/tradeoff. But current statements in the paragraph is really scattered with arguments on leaf-radiation relationship, canopy level process, species composition. It is not clear how those factors are linked to ecosystem-level coordination, or the relevant hypothesis on how we expect these factors can influence ecosystem level coordination.
4. L127-128. What is the implication of finding ecosystem level coordination? A dynamic vegetation model would generally simulate processes at leaf level first and then upscale to the ecosystem scale. Do you mean the ecosystem level coordination/tradeoff can be used to improve the upscale algorithm? Or they serve as a baseline to validate functional response from models. Need to be specific on the implications and meaning of the study.
5. L142. By saying “site-weighted”, I am guessing it really means species-weighted or community-weighted, as I cannot find in Methods how you assigned weight to different sites. In Method section, you used “community-weighted”, need to be consistent throughout.
6. L146. Here it goes back to the 1st comment, why we think 82% explanatory power of first two PCs is a good sign of coordination/tradeoff. A good option is to test some null hypotheses by generating random/unrelated traits records and see the explanatory power of first two PCs can be comparable to what you got. This may be more relevant to the results in Fig 2 and 3, where the first 2 PCs explain even less variance.
7. L154, interesting to see leaf longevity is counted as an ecosystem trait, did you consider the weight of different age groups for evergreen ecosystems?
8. L162. Not very clear why the evidence provided here support Nmass is a better indicator than Narea. I would argue since Nmass is dependent on LMA, their coordination is more or less expected, and may have enhanced PC you got. Narea would be a more independent from LMA, less likely to artificially enhance PC.
9. Why here there is a need to apply Dray’s method and do bootstrapping while for the Fig. 1 result there is no need. Does bootstrapping means PCs derived in each iteration are different? If it is, why?

10. "performance-persistence" means?

11. L189 and fig.2. Would it be a bit stretching to conclude PC1 represent maximum ecosystem processes, PC2 represents size effect, and PC3 represents structure effect. They seem to have clear overlaps. For example, in PC1, we also see considerable contributions from variables representing ecosystem size, such as LAI, LMA.

12. L231 and Fig 3, not very convincing to see PC1 represents maximum rate of processes, while PNUE has a negative loading. Not clear what PC2 represents, even it explains 29% of the variance. I am also curious how can we link these PCs to least cost, what indicate here is maybe just some tradeoffs.

13. L266-267, citation? Would be nice to compare the PCs from this analysis to PCs from studies on leaf level traits, to quantitatively show coordination/tradeoff differs between scales.

14. L272-273, I am not sure current result supports this interpretation. Please see comment 12.

15. L307. Not sure eddy covariance data photosynthetic/respiration capacity is independent of mass or area normalizing. Eddy covariance derived variables have unit of per m², and on m² there could be different amount of leaf masses. So if we use per mass for these variables, it is likely to generate different results.

16. L316. How to use the information from a modelling perspective. So at leaf-level simulation, LL increases then A decreases, however, after upscaling we should see LL increases then A increases?

17. L322-325. Since PC1 (leaf economic spectrum) and PC2 (canopy structure) are orthogonal to each other, does that mean leaf economic spectrum and canopy structure are unlikely to influence each other?

18. L358-360. Water is an important dimension when water-related variables are used in PCA...what is the message here?

19. L367. "we find similar component" in which part of the results?

Reviewer #2 (Remarks to the Author):

In this paper, three published individual-level plant trait economic spectrums are tested at the ecosystem (vegetation) level. Based on the similarity of the results at the different levels of organization it is concluded that "Leaf-level coordination principles propagate to the ecosystem scale". It is argued that this "...supports the development of more realistic global dynamic vegetation models with critical empirical data, reducing the uncertainty of climate change projections".

The paper will be of great interest to the plant-traits scientific community, especially the empirical scientists. The paper is well written, although readability could be improved by dividing the discussion into subsections, and better explaining the meaning of some terms, such as "eigenvectors", for non-PCA experts. However, in its current form it is not perfectly clear how the paper can support DGVM development and therefore it may be of limited interest to the modelling community. Below we make some suggestions on how the manuscript could be modified to become more useful for modellers.

The authors are after predictive principles for vegetation properties, but it would be good to be more specific about the terms used. The optimality principles giving rise to the trait economic spectrums at the individual level are the result of genetic evolution at the individual level (see e.g. Franklin et al. 2020). Although it makes sense in a statistical sense, the question "whether well-established

coordination principles that apply to the leaf and plant scales are conserved at the ecosystem scale” does not make sense from an eco-evolutionary perspective. Nothing is coordinated at the ecosystem level in the sense of evolution of coordination of processes towards improved fitness. In contrast, it is true that the coordination principles propagate to the ecosystem scale in the sense that the within-individual trait correlations explain a significant part of the trait correlations among communities and ecosystems. Thus, the question that is actually analyzed here is whether individual scale coordination principles can be used to approximate correlations among community mean traits and ecosystem processes.

The following is meant as suggestions and not as criticism or required revisions. There is an extensive discussion of scale emergent properties that are not explained by the individual-based spectrums, which is quite informative for ecologists. However, to make the study more relevant to dynamic vegetation models it would be useful with some additional explanation of underlying processes affecting the variables analyzed. Two different types of “ecosystem” variables are used, (i) community mean traits (e.g. LMA, Nmass) that are determined by individual traits and community composition, and (ii) ecosystem processes (e.g. GPP), which are additionally affected by plant density (or vegetation cover). In DGVMs you want to separate these processes, e.g. as plant density is directly affected by disturbances and management whereas community composition is not necessarily changed. Density is related to leaf area index, which is mentioned as a confounding factor (line 273). Such factors could be investigated as random effects in linear mixed effects models (for instance see: <https://cran.r-project.org/web/packages/lme4/vignettes/lmer.pdf>).

Another way to make the results more useful for DGVMs is to separate divergent functional types (PFTs; most DGVMs represent different PFTs). For instance, the trade-off depicted in the principal component analysis (Fig. 1) clearly separates different plant functional groups, such as needle-leaved gymnosperms (blue dots on the left side) and broad-leaved angiosperms (brown dots on the right side). Similarly, the trade-off depicted in the principal component analysis (Fig. 2) shows the obvious differences between forest and grassland species, and therefore should reflect differences in tissue investment (wSSD) and canopy height (Hc) between these ecosystems. Hence, further dissecting this dataset into the different ecosystem types (forests, grassland, etc.) and associated plant functional types (trees, grasses, etc.) may allow further insights. Having said that, the authors show that a similar analysis based on a subset of data conducted with only forests gave similar results but there is no comparison in statistical terms. The difference between statistical models could be tested by computing test statistics (with only forest and the full dataset) based on the Akaike information criterion (AIC).

Finally, we suggest that the authors make the dataset available, which would obviously simplify the further use of this work for model development.

Oskar Franklin & Florian Hofhansl, IIASA

Reference

Franklin, O., Harrison, S.P., Dewar, R., Farris, C.E., Brännström, Å., Dieckmann, U. et al. (2020). Organizing principles for vegetation dynamics. *Nat. Plants*, 6, 444-453.

Reviewer #1 (Remarks to the Author):

Dear reviewer,

Thank you for your time and very constructive comments. We have carefully read and included changes where we thought it was necessary. In cases where suggested changes were not made, we have explained why.

Specifically, we addressed all comments that highlighted confusing terms, unclear paragraphs, inaccurate wordings, or inadequately explained methodologies. We have also further discussed our interpretation or choice of methods in light of the constructive feedback received, and we have tested and discussed all the additional analyses suggested.

Below we report the point-by-point replies to the comments. The reviewer's comments are reported in bold-italics, our response is in normal text, and the parts added or changed in the manuscript are reported in italics between quotes, with corresponding line numbers.

In the manuscript “Leaf-level coordination principles propagate to the ecosystem scale”, Gomasasca et al. applied the principal component analysis (PCA) to multiple ecosystem variables derived from 98 flux sites. They tested whether the leaf economic spectrum, the global spectrum of form and function and the least cost hypothesis are valid at the ecosystem level, and concluded they are. The research question is well framed, and I can see it is natural follow-up to some earlier works from the group (Migliavacca et al. 2021). However, I found it was quite challenging to understand the meanings of each PCs and clearly grasp the implications of the study, in the current form of manuscript. I am afraid it still has a distance to go to reach the quality of nature communications.

1. While PCA is a useful method to identify the main type of factors that influence ecosystem functions, it is not necessarily clear to readers why higher explanatory power of first few PCs means there is traits coordination or tradeoff. Perhaps there is a need to clarify that in methods or introduction.

We have added an explanation of what each output in the PCA means in the Methods section:

“In the PCA results, the sign and direction of the eigenvectors denote relationships and trade-offs between the variables (arrows in the panels ‘a’ of Figures 1-3). Eigenvectors that are orthogonal to one another suggest trade-offs between the corresponding variables, while parallel eigenvectors indicate correlations. The same concept applies to the loadings, which represent the projections of the eigenvectors on each principal component: loadings with a different sign highlight potential trade-offs between variables, and equal signs indicate potential correlations (panels ‘c’ in Figures 1-3). A high explained variance assures that the selected variables are appropriately capturing the variance in the data for each principal component and as a whole (panels ‘b’ of Figures 1-3). Finally, the contributions describe how variables define each principal component (panels ‘d’ of Figures 1-3).” (L635-645)

We also added a similar paragraph at the beginning of the Results section, so that readers do not need to jump to the methods to gain insight into the meaning of the terms used:

“In our results, each variable is represented by eigenvectors that show their direction and strength in the hyperspace between principal components (Figure 1a). As in the leaf economic spectrum, we identified the key dimensions, or principal components, that explain more variance in the data (Figure 1b). Then, we assessed the projections of the eigenvectors (i.e. loadings) on the principal components and the relative contribution of each variable in defining each principal component.” (L151-156)

2. L105-106. I am not quite sure this is the only definition of the least cost theory, as some might have formulated the least cost hypothesis in a different way (e.g., Prentice 2014 Ecology Letters). As for water resource, I think the cost is not only about its “acquisition”, would also relate to the cost to prevent water from losing (especially in terms of water use efficiency).

We agree with the reviewer that there are more definitions, and we have updated our definition to more broadly describe variations of the least cost hypothesis, as well as to include the relevant aspect of water retention as suggested by the reviewer.

Now the revised manuscript reads:

“Another example of trait coordination is the least-cost hypothesis, which describes a continuum in plant economic strategies aimed at optimizing the input mix of two or more key limiting resources. The same economic theory can be applied to resource acquisition and utilization in plants: a decreasing acquisition and retention cost of one of two limiting resources (e.g., water) is generally accompanied by an increasing cost of the other limiting resource (e.g., nitrogen)^{7,10”} (L106-111)

3. L111-121. Do not quite follow here. I thought the paragraph is meant to introduce the concept of scale-emergent properties and then lead to few hypotheses on the existence of ecosystem level coordination/tradeoff. But current statements in the paragraph is really scattered with arguments on leaf-radiation relationship, canopy level process, species composition. It is not clear how those factors are linked to ecosystem-level coordination, or the relevant hypothesis on how we expect these factors can influence ecosystem level coordination.

The paragraph was indeed a bit hard to follow. We have restructured it to be more fluid. We start with introducing the complexity of ecosystems and the concept of scale-emergent properties, followed by an example of scale-emergent property, and ending with the take-away that leaf-level theory can contrast with the coordination of ecosystem-level properties:

“Ecosystems are intricate mixtures of different species that compete for resources such as energy, water, and nutrients¹², and abiotic drivers affecting biological processes and ecological interactions. Ecosystem-level processes are intrinsically linked to canopy architecture (e.g. arrangement of leaves, shoots, etc.)^{13,14}, and are determined by species composition, but are also influenced by disturbance and management. Consequently, ecosystems feature scale-emergent properties¹⁵⁻¹⁷, i.e. properties that are only manifested at a certain scale. For instance, light interception is largely dependent on canopy architecture due to the amount of light that can penetrate the canopy space; therefore, light response curves observed at the leaf level can be very different from the ones observed at the canopy scale¹⁸. In essence, the coordination between ecosystem functional properties at the canopy scale can contrast with the theory of optimization in leaves or plant organs¹⁹.” (L115-126)

4. L127-128. What is the implication of finding ecosystem level coordination? A dynamic vegetation model would generally simulate processes at leaf level first and then upscale to the ecosystem scale. Do you mean the ecosystem level coordination/tradeoff can be used to improve the upscale algorithm? Or they serve as a baseline to validate functional response from models. Need to be specific on the implications and meaning of the study.

We agree with the reviewer that this part should be clarified. We argue that both points raised by the reviewer are relevant implications of this work, and we clarified them now in the revised manuscript. Following the reviewer's suggestion, we added a new part:

“Evidence for ecosystem-level coordination would support the upscaling from simulated leaf-level processes to the ecosystem scale, while insight into scale-emergent properties could improve the upscaling algorithms used in dynamic vegetation models and serve to validate the functional response from models.” (L134-138).

This should clarify what the open question was (untested upscaling from leaf to ecosystem scale), and how our results have contributed to bridging this knowledge gap, while the rearrangement of the paragraph highlighted by the previous comment should aid readability and understanding:

“Ecosystems are intricate mixtures of different species that compete for resources such as energy, water, and nutrients¹², and abiotic drivers affecting biological processes and ecological interactions. Ecosystem-level processes are intrinsically linked to canopy architecture (e.g. arrangement of leaves, shoots, etc.)^{13,14}, and are determined by species composition, but are also influenced by disturbance and management. Consequently, ecosystems feature scale-emergent properties¹⁵⁻¹⁷, i.e. properties that are only manifested at a certain scale. For instance, light interception is largely dependent on canopy architecture due to the amount of light that can penetrate the canopy space; therefore, light response curves observed at the leaf level can be very different from the ones observed at the canopy scale¹⁸. In essence, the coordination between ecosystem functional properties at the canopy scale can contrast with the theory of optimization in leaves or plant organs¹⁹.” (L115-126)

Similarly, reviewer #2 asked for some additional work to clarify the implications for models. For instance, we used a mixed linear model to predict GPPsat and RECOmax (two typical model outputs) based on wLMA, wNmass, and wLL (three typical parameters) as fixed effects, and adding a random slope on the predictor, i.e. the (random) effect of LAImax (a state simulated by the models), for each vegetation class (IGBP; within which the parameters are fixed). This was fed to the model function as 'y ~ wNmass + wLMA + wLL + (LAImax | IGBP)' on a sample size of 87 sites. The results are now reported in Supplementary Table 5. We added a paragraph to the Conclusion subsection that reads:

"...in ecological theory and dynamic global vegetation models (DGVMs). DGVMs usually rely on constant vegetation parameters (e.g. mean traits) to simulate changes in carbon stocks (e.g. LAI) and ecosystem processes and fluxes. The DGVMs parameters are constant per plant functional type: for example, LMA or N content in leaves are parameterized as the mean values for large plant functional type classes such as deciduous, or evergreen forests. This parameterization typically neglects the variation in traits and the coordination between traits and functions observed in the literature. Instead, ecosystem functions (e.g. GPPsat, RECOmax) are simulated as response to foliage density (related to LAImax). This current paradigm can be not flexible enough to represent the variability and coordination between traits and functions and therefore can lead to biases in modelling³⁰. For instance, for the leaf economics spectrum, we can use a linear mixed model to test the relationship between GPPsat, the foliar traits (wLMA, wNmass, wLL), and the covariation between the variables once accounted for vegetation class and leaf area index as random effects. With this test we showed that some of the fixed effects resembling the trade-offs in the leaf economics spectrum at the ecosystem scales are important even when accounting for leaf area index and vegetation class, and should therefore not be overlooked (Supplementary Table 5). Recent studies focusing on DGVMs development are focusing on further including coordination principles, with explicit covariation of trait and functional parameters within vegetation cover classes²¹. In this sense, our analysis can help to indicate which traits and functions can be helpful in supporting the current developments." (L464-484)

Supplementary Table 5. Summary of linear mixed model calculated with the lmer function in the lme4² R Package, using the formula: $y \sim wNmass + wLMA + wLL + (LAI_{max} | IGBP)$, where y stands for the predicted variable (GPPsat or RECOmax), on 87 observations (sites).

GPPsat prediction						
Random effects						
Groups	Name	Variance	Std.Dev.	Corr		
IGBP (n = 10)	(Intercept)	0.53	0.72			
	LAI _{max}	1.90	1.38	1.00		
	Residual	25.30	5.03			
Fixed effects						
				Anova (Type II Wald chisquare tests)		
	Estimate	Std. Error	t value	Chisq	Df	Pr(>Chisq)
	(Intercept)	10.63	3.60	2.95		

wNmass	6.07	1.72	3.52	12.38	1	0.000
wLMA	-13.44	11.30	-1.19	1.42	1	0.234
wLL	-0.06	0.03	-1.65	2.72	1	0.099
Correlation of Fixed Effects						
	(Intr)	wNmass	wLMA			
wNmass	-0.80					
wLMA	-0.58	0.17				
wLL	-0.27	0.26	-0.31			
RECOmax prediction						
Random effects						
Groups	Name	Variance	Std.Dev.	Corr		

IGBP (n = 10)	(Intercept)	0.42	0.65			
	LAlmax	0.02	0.14	1.00		
	Residual	2.78	1.67			
Fixed effects						
				Anova (Type II Wald chisquare tests)		
	Estimate	Std. Error	t value	Chisq	Df	Pr(>Chisq)
(Intercept)	4.71	1.19	3.96			
wNmass	1.34	0.54	2.47	6.09	1	0.014
wLMA	-11.51	3.64	-3.16	9.99	1	0.002
wLL	0.00	0.01	-0.11	0.01	1	0.915
Correlation of Fixed Effects						

	(Intr)	wNmass	wLMA			
wNmass	-0.83					
wLMA	-0.58	0.23				
wLL	-0.27	0.26	-0.26			

5. L142. By saying “site-weighted”, I am guessing it really means species-weighted or community-weighted, as I cannot find in Methods how you assigned weight to different sites. In Method section, you used “community-weighted”, need to be consistent throughout.

We agree with the reviewer that this description might be confusing. We have changed this wording to be consistent throughout the manuscript and the supplementary information: we have kept the more broadly used formulation of “community-weighted”.

6. L146. Here it goes back to the 1st comment, why we think 82% explanatory power of first two PCs is a good sign of coordination/tradeoff. A good option is to test some null hypotheses by generating random/unrelated traits records and see the explanatory power of first two PCs can be comparable to what you got. This may be more relevant to the results in Fig 2 and 3, where the first 2 PCs explain even less variance.

Regarding the first part of the comment, 82% of explained variance is not a sign of coordination. It merely means that the first two PCs explain the vast majority of the variance. The coordination is explained by the direction and strength of the loadings. To avoid misleading formulations, we have changed the wording and sequence in this paragraph, which now reads:

“Two retained principal components (PC) cumulatively explained 82.3 ± 4.7 % of the variance in the dataset (Figure 1). The ecosystem-scale economics spectrum was apparent from the loadings of the principal component analysis (Figure 1c).” (L159-162)

Additionally, to avoid potential confusion, we clarified the terms used for the PCA results and their meaning, by adding an explanation of the PCA output at the beginning of the results, and in the methods section:

“In our results, each variable is represented by eigenvectors that show their direction and strength in the hyperspace between principal components (Figure 1a). As in the leaf economic spectrum, we identified the key dimensions, or principal components, that explain more variance in the data (Figure 1b). Then, we assessed the projections of the eigenvectors (i.e. loadings) on the principal components and the relative contribution of each variable in defining each principal component.” (L151-156)

“In the PCA results, the sign and direction of the eigenvectors denote relationships and trade-offs between the variables (arrows in the panels ‘a’ of Figures 1-3). Eigenvectors that are orthogonal to one another suggest trade-offs between the corresponding variables, while parallel eigenvectors indicate correlations. The same concept applies to the loadings, which represent the projections of the eigenvectors on each principal component: loadings with a different sign highlight potential trade-offs between variables, and equal signs indicate potential correlations (panels ‘c’ in Figures 1-3). A high explained variance assures that the selected variables are appropriately capturing the variance in the data for each principal component and as a whole (panels ‘b’ of Figures 1-3). Finally, the contributions describe how variables define each principal component (panels ‘d’ of Figures 1-3).” (L635-645)

Based on the reviewer’s suggestion “to test the null hypothesis”, we generated a dataset with random values for each variable. The simulated variables are normally distributed and independent of each other with mean and standard deviation determined by the actual data. This assumes that each property evolves independently, while extreme properties are selected against during evolution¹. We generated n = 499 datasets for each section of the analysis. We

repeated the PCA for each of the $n = 499$ datasets, and calculated the mean explained variance for each PC. With this test, we evaluated the explained variance by each of the PCs using randomized variables and compare it with the variance explained using the dataset. The null hypothesis is true when the explained variance of the PCs based on the actual dataset is statistically the same as the explained variance of the PCs based on the random dataset.

As seen below, the null hypothesis is rejected for the first PCs in each analysis. Borderline cases are PC2 in the leaf economics spectrum section ($23.1\% \pm 2.7\%$ for the actual dataset, $22.4\% \pm 1.0\%$ for the random dataset), and PC3 in the global spectrum section ($18.7\% \pm 1.9\%$ for the actual dataset, $17.5\% \pm 0.8\%$ for the random dataset). In these cases, the uncertainty in the estimates makes it hard to statistically distinguish the explained variances between the actual and random data. Since these PCs are the ones that we focus the least on in our interpretation, we prefer to avoid including this additional analysis in the manuscript.

Specifically, see below our results related to the first part (left, the leaf economics spectrum at the ecosystem scale, Figure 1b) compared to the results based on random data (right):

Below, our results related to the second part (left, the global spectrum of plant form and function at the ecosystem scale, Figure 2b), compared to the results based on random data (right):

And finally, our results related to the third part (left, the least cost hypothesis at the ecosystem scale, Figure 3b) compared to the results based on random data (right):

7. L154, interesting to see leaf longevity is counted as an ecosystem trait, did you consider the weight of different age groups for evergreen ecosystems?

Including different age groups of needles would definitely be interesting and improve our analysis. Unfortunately, we have yet to find this information for many sites in a consistent way, and leaf longevity data is already very limiting. So, we had to settle for maximum leaf lifespan. Although not optimal, this is so far the only compromise that enables the inclusion of leaf longevity estimates in the analysis. We have stated this in the methods section:

“Regarding leaf longevity, we could not account for different leaf age groups because of a lack of data.” (L549-550)

8. L162. Not very clear why the evidence provided here support Nmass is a better indicator than Narea. I would argue since Nmass is dependent on LMA, their coordination is more or less expected, and may have enhanced PC you got. Narea would be a more independent from LMA, less likely to artificially enhance PC.

There are multiple reasons why we chose to include Nmass + LMA instead of Narea, and we have now clarified the following in the text.

First and most importantly, the original formulation of the Leaf Economics Spectrum by Wright et al. (2004) used Nmass + LMA, and we wanted to be as comparable as possible to that study. This is explained in the discussion of the results:

“Here, we used mass-based traits to be coherent with the leaf economics spectrum.” (L195-196)

Second, our nitrogen measurements are done directly at the sites on a mass basis. Converting those measurements to Narea would require the LMA, and this would exclude the LMA metric from the analysis.

Third, our results of multi-model inference showed not only better R^2 , but also significantly better mean model performance when including both wNmass and wLMA (AICc - Akaike’s Information Criterion = 551.5) instead of wNarea (AICc = 568.7). This was not shown, so we have added the values of model performance in Supplementary Fig. 2. Note that a lower AICc denotes comparatively better model performance.

For the above-mentioned reasons, we prefer to keep the analysis as they are, but we have included the results of the first PCA on the leaf economics spectrum analogue based on wNarea in this response. Since wNarea is calculated based on wLMA, its loading is negative.

9. Why here there is a need to apply Dray's method and do bootstrapping while for the Fig. 1 result there is no need. Does bootstrapping means PCs derived in each iteration are different? If it is, why?

The methodology applied is the same for all sections of the analysis, including for Figure 1. The Dray method was used in all three sections to select the number of significant principal components. The only difference here is that we decided to avoid showing all six PCs in Figure 2 for the sake of clarity and better visualization (but all six PCs are shown in the Supplementary Table 3). While the Dray method was applied to identify the number of components to be retained, the bootstrapping was performed to gain insight into the uncertainty of explained variances, loadings, and contributions in the PCA output. So yes, this means that the PCs derived in each bootstrapping iteration are slightly different, because a different input is fed to the PCA. However, note that this information was only used to derive the uncertainties, while the results shown in the figures are based on the full dataset without bootstrapping, to avoid losing

information. We slightly modified the Methods section and the wording in the Results to better explain these points and avoid confusion:

“Six significant principal components were retained based on the Dray method²², (Supplementary Table 3). However, we concentrate our interpretation to the first three principal components [...]” (L230-231)

10. “performance-persistence” means?

We designate the trade-off in the leaf economics spectrum as “performance-persistence trade-off”. To make this more understandable, we have added this more formally in the introduction, after the description of the leaf economics spectrum. Thereafter, it should be clearer what we refer to with this formulation:

“We refer to this as the performance-persistence trade-off, because resource acquisition costs can either be directed towards resource conservation and leaf persistence¹⁰, or fast growth and photosynthetic performance.” (L100-102).

11. L189 and fig.2. Would it be a bit stretching to conclude PC1 represent maximum ecosystem processes, PC2 represents size effect, and PC3 represents structure effect. They seem to have clear overlaps. For example, in PC1, we also see considerable contributions from variables representing ecosystem size, such as LAI, LMA.

The eigenvectors indeed have contributions on more than one principal component. This is true by definition, but it is particularly apparent for PC1 and PC2 in Figure 2, since the two described axis of leaf economics (wNmass, wLMA) and ecosystem size (LAI_{max}, Hc) are rotated nearly 45° relative to PC1 and PC2. This is what was described in the global spectrum of plant form and function by Díaz et al. (2016), and what we explored in the discussion. Indeed, one way to interpret PCA results is to look at the relationship between eigenvectors, rather than their projections on the principal components (loadings and contributions). However, we feel that disregarding loadings and contributions would only convey part of the informative potential provided by the PCA output. This is why we decided to focus more on describing the projections

on the PCs. This way, overlaps will be often unavoidable, but we argue that the main properties are still visible on each PC. The fact that size and structural elements contribute to PC1 further suggests that productivity is affected by increased vegetation biomass. The leaf economics spectrum represented by wNmass and wLMA is a trade-off between photosynthetic performance and structural persistence, and, accordingly, the contributions of wNmass and wLMA are equally distributed between PC1 and PC2. PC3 and beyond are more problematic to interpret, but we tried to explore the potential reasons behind our results in the discussion.

Based on the above, we have added a new paragraph that should clarify some of the concerns raised:

“The fact that size and structural elements such as leaf area index and canopy height also contribute to PC1 highlights how ecosystem processes are affected by increased vegetation biomass. The leaf economics spectrum represented by wNmass and wLMA is a trade-off between photosynthetic performance and structural persistence. Accordingly, the contributions of wNmass and wLMA are equally distributed between PC1 and PC2.” (L259-264).

12. L231 and Fig 3, not very convincing to see PC1 represents maximum rate of processes, while PNUE has a negative loading. Not clear what PC2 represents, even it explains 29% of the variance. I am also curious how can we link these PCs to least cost, what indicate here is maybe just some tradeoffs.

We agree that in the results section it is hard to argue that PC1 represents the maximum rates based on the variables selected, since not all the variables directly represent maximum rates, and PNUE is negative (although hardly relevant on PC1). However, the fact that variables such as Hc, Gsmax, EF, and WUEt have positive loadings on PC1 should indicate that this axis reflects the gradient between low-stature vegetation with limited available resources (e.g. water availability, or cold temperatures) with resulting low maximum surface conductance to high stature vegetation with high water availability. The fact that PNUE is inversely related makes sense in this context as even while the photosynthetic capacity (GPPsat) increases along this gradient, photosynthesis per unit of Nitrogen is instead decreasing following the least cost hypothesis. For

instance, adding GPPsat to the Least Cost Hypothesis analysis makes PC1 clearly related to maximum rates. Moreover, in previous studies (Migliavacca et al., 2021) the gradient of short to tall vegetation was clearly associated with a gradient of process rates (from low maximum basal respiration, photosynthetic capacity, and surface conductance, to high basal respiration, photosynthetic capacity and surface conductance). To expand on these points, we have included some additional explanations in the text as shown below.

“This was consistent when including GPPsat or different metrics related to productivity and photosynthetic nitrogen use efficiency (e.g. Supplementary Fig. 4, Supplementary Fig. 5). The fact that variables such as canopy height, maximum surface conductance, evaporative fraction, and water use-efficiency have positive loadings on PC1 indicates that this component reflects the gradient between low stature vegetation with limited available resources and low maximum surface conductance (e.g. water limited, or low temperature) to high stature vegetation with high water availability. The effect of PNUE is inversely related to the other variables, in line with the least cost hypothesis, but hardly relevant on PC1 (Figure 3). The dimension of “maximum rates” consistently emerges at the ecosystem scale, regardless of the set of chosen variables (Supplementary Fig. 4, Supplementary Fig. 5).” (L342-351)

We argue that PC2 represents the trade-offs underlying the least-cost hypothesis, as we presented and discussed at the lines highlighted below:

“The second principal component explained 29 ± 1.8 % of the variance (Figure 3b) and uncovered the trade-offs expected by the least-cost hypothesis: a negative relationship between the loadings of water use-efficiency and canopy height on one side (WUEt: -0.54 ± 0.55 , and Hc: -0.39 ± 0.51), and evaporative fraction, photosynthetic nitrogen use-efficiency, and surface conductance on the other side (EF: 0.74 ± 0.32 , PNUE: 0.65 ± 0.47 , and Gsmax: 0.54 ± 0.26 , Figure 3c and 3d, Supplementary Table 4).” (L352-357)

“Our results also show the negative relationship between PNUE and WUEt on the second dimension of the principal component analysis, as expected from leaf-level field studies and theory^{2,7,11,35–37}. Additionally, other expected trade-offs are present on this component, such as

the negative relationship between surface conductance and water use-efficiency³⁸, or a negative relationship between WUEt and evaporative fraction, which is low in more arid sites and higher in wet sites. This is in line with the expected increase in the efficiency of plants in using water along aridity gradients, as shown with leaf-level measurements of leaf-internal to ambient CO₂ ratio as a proxy of intrinsic water use efficiency^{39,40}. In sum, the second component in our third and final analysis unravels the axis of the least-cost hypothesis. The coordination between the variables of the least-cost hypothesis covers a range of sites from wet conditions with high efficiency of photosynthetic nitrogen use, but low water use-efficiency, to arid conditions with high efficiency of water use, but low photosynthetic nitrogen use-efficiency.” (L374-386)

In particular, we tried to incorporate all the variables related both to the Wright et al., 2003 formulation, and the Prentice et al., 2014 formulation, at least for the variables that we could access at this scale. We considered multiple aspects of the different definitions (e.g. PNUE vs WUE, WUE vs surface conductance, WUE vs evaporative fraction, a measure of aridity). It is clear that with more data such as leaf-internal and ambient CO₂ mole fraction, or stable carbon isotope signatures measured at the sites, we could have better explained the mechanisms. However, these data are not consistently collected at the sites. Therefore, we stopped the analysis at the level of the emergent trade-offs expected from the theory between PNUE and WUEt, G_{max}, and EF. To underline this limitation, we added a sentence to the discussion, which reads as follows:

“Measures of leaf-internal and ambient CO₂ mole fraction, or stable carbon isotope signatures measured at the sites would help to strengthen our claims related to the least cost hypothesis, but these measurements were unavailable for the large majority of sites.” (L386-389)

13. L266-267, citation? Would be nice to compare the PCs from this analysis to PCs from studies on leaf level traits, to quantitatively show coordination/tradeoff differs between scales.

Citations added for the leaf economics spectrum and global spectrum of plant form and function (L456).

Comparison of PCA results are only possible with these two studies. However, direct, quantitative comparison might not be possible, since we do not have the exact same sets of variables (e.g. we

are missing phosphorous estimates for the leaf economics spectrum, or seed mass for the global spectrum). We have tried to add those variables suggested by the reviewer at the beginning of the analysis, but that resulted in reduced sample sizes and thus reduced robustness of the analysis. For this reason, certain variables were excluded and we did not proceed with a direct quantitative comparison between leaf level and ecosystem level relationships. Rather, we discuss qualitatively the relationships and the similarities or differences between leaf level literature results and ecosystem level results.

14. L272-273, I am not sure current result supports this interpretation. Please see comment 12.

We have changed the wording of this sentence to be more general:

“e.g. optimization of nitrogen use and water use is a secondary dimension” (L461-462).

Additionally, we have expanded parts of the results and discussion to better present these points.

“This was consistent when including GPPsat or different metrics related to productivity and photosynthetic nitrogen use efficiency (e.g. Supplementary Fig. 4, Supplementary Fig. 5).” (L342-343)

“The fact that variables such as canopy height, maximum surface conductance, evaporative fraction, and water use-efficiency have positive loadings on PC1 indicate that this component reflects the gradient between low stature vegetation with limited available resources with resulting low maximum surface conductance (e.g. water limited, or low temperature) to high stature vegetation with high water availability. The effect of PNUE is inversely related to the other variables, in line with the least cost hypothesis, but hardly relevant on PC1 (Figure 3). The dimension of “maximum rates” consistently emerges at the ecosystem scale, regardless of the set of chosen variables (Supplementary Fig. 4, Supplementary Fig. 5).” (L344-351)

“Measures of leaf-internal and ambient CO₂ mole fraction, or stable carbon isotope signatures measured at the sites would help to strengthen our claims related to the least cost hypothesis, but these measurements were unavailable for the large majority of sites.” (L386-389)

15. L307. Not sure eddy covariance data photosynthetic/respiration capacity is independent of mass or area normalizing. Eddy covariance derived variables have unit of per m2, and on m2 there could be different amount of leaf masses. So if we use per mass for these variables, it is likely to generate different results.

Indeed, the sentence “photosynthetic capacity and respiration are estimated from eddy covariance measurements and are thus independent of leaf mass- (or area-) normalization.” is not correct as the eddy covariance quantities are measured by area and the measured values depend also by the leaf area index of the ecosystem observed. So, we thank the reviewer for spotting this erroneous sentence that we have now removed from the text.

Instead of calculating the fluxes normalized by leaf area index, we decided to use LAImax in the multivariate analysis. This is why we included LAImax in the subsequent section of the analysis and in the predictive models of Supplementary Fig. 2. See below an output with GPPsat normalized by LAImax. Note that the relationships of the leaf economics spectrum are still present, but with higher uncertainty.

For the LAI normalization, we could have used satellite data. However, firstly, satellite LAI retrieved is typically saturating at high values (e.g. MODIS saturated above 5-6 $\text{m}^2 \text{m}^{-2}$) and, secondly, there is a mismatch between eddy covariance footprint and satellite pixels. These issues can generate uncertainties in the normalization of fluxes by LAI. Therefore, we used LAI_{max} only in the multivariate analysis and as stand-alone variable in the PCA.

16. L316. How to use the information from a modelling perspective. So at leaf-level simulation, LL increases then A decreases, however, after upscaling we should see LL increases then A increases?

This comment greatly synergizes with the suggestions from reviewer 2, which we have included in the revised manuscript. To specifically answer the question here, we show that the same principles valid for the leaf level are found at the ecosystem scale, which means that a “bigleaf” modeling approach seems to be an acceptable approximation to represent these trade-offs. So, at the ecosystem level, as well as at the leaf level, A (GPP_{sat}) decreases with increasing LL (wLL), which can be seen in PC1 of Figure 1c. However, this trade-off at the ecosystem level is not the whole story: additional properties (scale-emergent) need to be considered, which is what we see on PC2. The subsequent analyses that include structure and hydraulics are our attempt at describing such additional scale-emergent properties. Additionally, dynamic plant traits seem important when testing a simple linear mixed model.

Some of the additions to the text are as follows, and the output of the linear mixed model was added in the supplementary information (Supplementary Table 5):

“...in ecological theory and dynamic global vegetation models (DGVMs). DGVMs usually rely on constant vegetation parameters (e.g. mean traits) to simulate changes in carbon stocks (e.g. LAI) and ecosystem processes and fluxes. The DGVMs parameters are constant per plant functional type: for example, LMA or N content in leaves are parameterized as the mean values for large plant functional type classes such as deciduous, or evergreen forests. This parameterization typically neglects the variation in traits and the coordination between traits and functions observed in the literature. Instead, ecosystem functions (e.g. GPP_{sat}, RECO_{max}) are simulated as

response to foliage density (related to LAImax). This current paradigm can be not flexible enough to represent the variability and coordination between traits and functions and therefore can lead to biases in modelling³⁰. For instance, for the leaf economics spectrum, we can use a linear mixed model to test the relationship between GPPsat, the foliar traits (wLMA, wNmass, wLL), and the covariation between the variables once accounted for vegetation class and leaf area index as random effects. With this test we showed that some of the fixed effects resembling the trade-offs in the leaf economics spectrum at the ecosystem scales are important even when accounting for leaf area index and vegetation class, and should therefore not be overlooked (Supplementary Table 5). Recent studies focusing on DGVMs development are focusing on further including coordination principles, with explicit covariation of trait and functional parameters within vegetation cover classes²¹. In this sense, our analysis can help to indicate which traits and functions can be helpful in supporting the current developments.” (L464-484)

Supplementary Table 5. Summary of linear mixed model calculated with the lmer function in the lme4² R Package, using the formula: $y \sim wNmass + wLMA + wLL + (LAImax | IGBP)$, where y stands for the predicted variable (GPPsat or RECOmax), on 87 observations (sites).

GPPsat prediction						
Random effects						
Groups	Name	Variance	Std.Dev.	Corr		
IGBP (n = 10)	(Intercept)	0.53	0.72			
	LAImax	1.90	1.38	1.00		

	Residual	25.30	5.03			
Fixed effects						
				Anova (Type II Wald chisquare tests)		
	Estimate	Std. Error	t value	Chisq	Df	Pr(>Chisq)
(Intercept)	10.63	3.60	2.95			
wNmass	6.07	1.72	3.52	12.38	1	0.000
wLMA	-13.44	11.30	-1.19	1.42	1	0.234
wLL	-0.06	0.03	-1.65	2.72	1	0.099
Correlation of Fixed Effects						
	(Intr)	wNmass	wLMA			
wNmass	-0.80					

wLMA	-0.58	0.17				
wLL	-0.27	0.26	-0.31			
RECOmax prediction						
Random effects						
Groups	Name	Variance	Std.Dev.	Corr		
IGBP (n = 10)	(Intercept)	0.42	0.65			
	LAlmax	0.02	0.14	1.00		
	Residual	2.78	1.67			
Fixed effects						
				Anova (Type II Wald chisquare tests)		
	Estimate	Std. Error	t value	Chisq	Df	Pr(>Chisq)

(Intercept)	4.71	1.19	3.96			
wNmass	1.34	0.54	2.47	6.09	1	0.014
wLMA	-11.51	3.64	-3.16	9.99	1	0.002
wLL	0.00	0.01	-0.11	0.01	1	0.915
Correlation of Fixed Effects						
	(Intr)	wNmass	wLMA			
wNmass	-0.83					
wLMA	-0.58	0.23				
wLL	-0.27	0.26	-0.26			

17. L322-325. Since PC1 (leaf economic spectrum) and PC2 (canopy structure) are orthogonal to each other, does that mean leaf economic spectrum and canopy structure are unlikely to influence each other?

This is a good question. We would rather say these sets of variables (wNmass and wLMA vs LAImax) are not correlated, and possibly only indirectly linked. However, notice how the eigenvector of GPPsat (arguably the response variable) lies between LAImax and wNmass in Figure 2a. This suggests that both the leaf economics spectrum (wNmass and wLMA) and vegetation size (LAImax and to a minor degree Hc) concurrently influence GPPsat, as we described at the lines below:

“This suggests that the leaf economics spectrum and the size dimension of vegetation (LAImax and Hc) combined likely explain the photosynthetic performance of the ecosystems.” (L255-256)

18. L358-360. Water is an important dimension when water-related variables are used in PCA...what is the message here?

Thank you for spotting this inconsistency. Indeed, this sentence is unclear. We rephrased it into:

“In the following section related to the least-cost hypothesis we show that the dimension related to water is indeed important.” (L297-299).

19. L367. “we find similar component” in which part of the results?

We clarified this part by adding:

“[...] in our analogue of the global spectrum of plant form and function” (L306-307)

Reviewer #2 (Remarks to the Author):

Dear reviewers,

Thank you for your time, positive feedback and very constructive comments. We have carefully read and tried to implement as many suggestions as possible. In case suggested changes were not made we have explained why.

Specifically, we have tried to improve readability by better explaining confusing terms, unclear paragraphs, or inaccurate wordings. We have also tested and discussed the additional analyses suggested, where possible based on data constraints.

Below we report the point-by-point replies to the comments. The reviewer's comments are reported in bold-italics, our response is in normal text, and the parts added or changed in the manuscript are reported in italics between quotes, with corresponding line numbers.

In this paper, three published individual-level plant trait economic spectrums are tested at the ecosystem (vegetation) level. Based on the similarity of the results at the different levels of organization it is concluded that “Leaf-level coordination principles propagate to the ecosystem scale”. It is argued that this “...supports the development of more realistic global dynamic vegetation models with critical empirical data, reducing the uncertainty of climate change projections”.

The paper will be of great interest to the plant-traits scientific community, especially the empirical scientists. The paper is well written, although readability could be improved by dividing the discussion into subsections, and better explaining the meaning of some terms, such as “eigenvectors”, for non-PCA experts. However, in its current form it is not perfectly clear how the paper can support DGVM development and therefore it may be of limited interest to the modelling community. Below we make some suggestions on how the manuscript could be modified to become more useful for modellers.

Regarding the first point, we agree that dividing the Discussion into subsections would improve readability, but this clashes with the formatting instructions of Nature Communications. We decided to move the three subsections of the Discussion to the corresponding subsections of the

Results, thus streamlining the text and hopefully avoiding unnecessary and confusing jumps between results and discussion. This way, the 'Results and discussion' are now divided into three clearly distinct subsections that present and discuss the outcomes of the analyses, and a 'Conclusion' subsection with synthetic statements about the main findings, the relevance of the study, and the outlook into future research questions.

Moreover, we added some explanations of the PCA method at the start of the results and in the methods section. These should clarify the meaning of the technical terms to a broader audience.

"In our results, each variable is represented by eigenvectors that show their direction and strength in the hyperspace between principal components (Figure 1a). As in the leaf economic spectrum, we identified the key dimensions, or principal components, that explain more variance in the data (Figure 1b). Then, we assessed the projections of the eigenvectors (i.e. loadings) on the principal components and the relative contribution of each variable in defining each principal component." (L151-156)

"In the PCA results, the sign and direction of the eigenvectors denote relationships and trade-offs between the variables (arrows in the panels 'a' of Figures 1-3). Eigenvectors that are orthogonal to one another suggest trade-offs between the corresponding variables, while parallel eigenvectors indicate correlations. The same concept applies to the loadings, which represent the projections of the eigenvectors on each principal component: loadings with a different sign highlight potential trade-offs between variables, and equal signs indicate potential correlations (panels 'c' in Figures 1-3). A high explained variance assures that the selected variables are appropriately capturing the variance in the data for each principal component and as a whole (panels 'b' of Figures 1-3). Finally, the contributions describe how variables define each principal component (panels 'd' of Figures 1-3)." (L635-645)

The second part of the comments is addressed point by point below.

The authors are after predictive principles for vegetation properties, but it would be good to be more specific about the terms used. The optimality principles giving rise to the trait economic spectrums at the individual level are the result of genetic evolution at the individual level (see

e.g. Franklin et al. 2020). Although it makes sense in a statistical sense, the question “whether well-established coordination principles that apply to the leaf and plant scales are conserved at the ecosystem scale” does not make sense from an eco-evolutionary perspective. Nothing is coordinated at the ecosystem level in the sense of evolution of coordination of processes towards improved fitness. In contrast, it is true that the coordination principles propagate to the ecosystem scale in the sense that the within-individual trait correlations explain a significant part of the trait correlations among communities and ecosystems. Thus, the question that is actually analyzed here is whether individual scale coordination principles can be used to approximate correlations among community mean traits and ecosystem processes.

Thanks for this comment. Indeed, we referred to optimization in a functional sense, and not from a strictly evolutionary perspective. Therefore, we are looking more at what is suggested by the reviewer: how “*individual scale coordination principles can be used to approximate correlations among community mean traits and ecosystem processes*”. For this reason, we have changed the formulation of the hypotheses based on the suggestions by the reviewers:

“Here, we ask whether well-established coordination principles that apply to the leaf and plant scales can be used to approximate ecosystem-scale coordination among community mean traits and ecosystem processes” (L139-141)

The following is meant as suggestions and not as criticism or required revisions. There is an extensive discussion of scale emergent properties that are not explained by the individual-based spectrums, which is quite informative for ecologists. However, to make the study more relevant to dynamic vegetation models it would be useful with some additional explanation of underlying processes affecting the variables analyzed. Two different types of “ecosystem” variables are used, (i) community mean traits (e.g. LMA, Nmass) that are determined by individual traits and community composition, and (ii) ecosystem processes (e.g. GPP), which are additionally affected by plant density (or vegetation cover). In DGVMs you want to separate these processes, e.g. as plant density is directly affected by disturbances and management whereas community composition is not necessarily changed. Density is related to leaf area index, which is mentioned as a confounding factor (line 273). Such factors could be investigated

as random effects in linear mixed effects models (for instance see: <https://cran.r-project.org/web/packages/lme4/vignettes/lmer.pdf>).

We used the mixed linear model suggested by the reviewers to predict GPPsat and RECOmax based on wLMA, wNmass, and wLL as fixed effects, and adding a random slope on the predictor, i.e. the (random) effect of LAImax, for each vegetation class (IGBP). This was fed to the model function as 'y ~ wNmass + wLMA + wLL + (LAImax | IGBP)' on a sample size of 87 sites.

When predicting GPPsat, the fixed effects were -0.06 for wLL, -13.44 for wLMA, and 6.07 for wNmass, but only the effect of wNmass was considered significant based on p-values in anova (although p-values are criticized by the authors of the linear mixed model package). The within group (IGBP) standard deviation of LAImax was 1.38. When predicting RECOmax, the fixed effects were near 0 for wLL, -11.51 for wLMA, and 1.34 for wNmass, with both wLMA and wNmass being significant based on anova results. The within group (IGBP) standard deviation of LAImax was 0.14.

In both cases, we see that the direction of the relationships expected by the leaf economics spectrum is retained for wLMA and wNmass, mostly in a significant way based on ANOVA results (but again, this method might not be best suited in this context, and even non-significant effects should not be disregarded). The random effect of LAImax was important for GPPsat, and hardly relevant for respiration, which makes sense based on the fact that respiration measurements are taken for the whole system, but only photosynthetic tissues (i.e. leaves, related to LAI) define the photosynthetic capacity.

We argue that including these results in the main text would go beyond the original scope of our manuscript, but we added the output summary table in the supplementary information (Supplementary Table 5), and referenced in the text accordingly:

"...in ecological theory and dynamic global vegetation models (DGVMs). DGVMs usually rely on constant vegetation parameters (e.g. mean traits) to simulate changes in carbon stocks (e.g. LAI) and ecosystem processes and fluxes. The DGVMs parameters are constant per plant functional type: for example, LMA or N content in leaves are parameterized as the mean values for large

plant functional type classes such as deciduous, or evergreen forests. This parameterization typically neglects the variation in traits and the coordination between traits and functions observed in the literature. Instead, ecosystem functions (e.g. GPPsat, RECOmax) are simulated as response to foliage density (related to LAImax). This current paradigm can be not flexible enough to represent the variability and coordination between traits and functions and therefore can lead to biases in modelling³⁰. For instance, for the leaf economics spectrum, we can use a linear mixed model to test the relationship between GPPsat, the foliar traits (wLMA, wNmass, wLL), and the covariation between the variables once accounted for vegetation class and leaf area index as random effects. With this test we showed that some of the fixed effects resembling the trade-offs in the leaf economics spectrum at the ecosystem scales are important even when accounting for leaf area index and vegetation class, and should therefore not be overlooked (Supplementary Table 5). Recent studies focusing on DGVMs development are focusing on further including coordination principles, with explicit covariation of trait and functional parameters within vegetation cover classes²¹. In this sense, our analysis can help to indicate which traits and functions can be helpful in supporting the current developments.” (L464-484)

Supplementary Table 5. Summary of linear mixed model calculated with the lmer function in the lme4² R Package, using the formula: $y \sim wNmass + wLMA + wLL + (LAImax | IGBP)$, where y stands for the predicted variable (GPPsat or RECOmax), on 87 observations (sites).

GPPsat prediction						
Random effects						
Groups	Name	Variance	Std.Dev.	Corr		
IGBP (n = 10)	(Intercept)	0.53	0.72			

	LAlmax	1.90	1.38	1.00		
	Residual	25.30	5.03			
Fixed effects						
				Anova (Type II Wald chisquare tests)		
	Estimate	Std. Error	t value	Chisq	Df	Pr(>Chisq)
(Intercept)	10.63	3.60	2.95			
wNmass	6.07	1.72	3.52	12.38	1	0.000
wLMA	-13.44	11.30	-1.19	1.42	1	0.234
wLL	-0.06	0.03	-1.65	2.72	1	0.099
Correlation of Fixed Effects						
	(Intr)	wNmass	wLMA			

wNmass	-0.80					
wLMA	-0.58	0.17				
wLL	-0.27	0.26	-0.31			
RECOmax prediction						
Random effects						
Groups	Name	Variance	Std.Dev.	Corr		
IGBP (n = 10)	(Intercept)	0.42	0.65			
	LAlmax	0.02	0.14	1.00		
	Residual	2.78	1.67			
Fixed effects						
				Anova (Type II Wald chisquare tests)		

	Estimate	Std. Error	t value	Chisq	Df	Pr(>Chisq)
(Intercept)	4.71	1.19	3.96			
wNmass	1.34	0.54	2.47	6.09	1	0.014
wLMA	-11.51	3.64	-3.16	9.99	1	0.002
wLL	0.00	0.01	-0.11	0.01	1	0.915
Correlation of Fixed Effects						
	(Intr)	wNmass	wLMA			
wNmass	-0.83					
wLMA	-0.58	0.23				
wLL	-0.27	0.26	-0.26			

Another way to make the results more useful for DGVMs is to separate divergent functional types (PFTs; most DGVMs represent different PFTs). For instance, the trade-off depicted in the

principal component analysis (Fig. 1) clearly separates different plant functional groups, such as needle-leaved gymnosperms (blue dots on the left side) and broad-leaved angiosperms (brown dots on the right side). Similarly, the trade-off depicted in the principal component analysis (Fig. 2) shows the obvious differences between forest and grassland species, and therefore should reflect differences in tissue investment (wSSD) and canopy height (Hc) between these ecosystems. Hence, further dissecting this dataset into the different ecosystem types (forests, grassland, etc.) and associated plant functional types (trees, grasses, etc.) may allow further insights. Having said that, the authors show that a similar analysis based on a subset of data conducted with only forests gave similar results but there is no comparison in statistical terms. The difference between statistical models could be tested by computing test statistics (with only forest and the full dataset) based on the Akaike information criterion (AIC).

Since the principal component analysis is a dimensionality reduction method, and not a predictive model analysis, no AIC metric is available. At the same time, dissecting the dataset into subgroups can be problematic, since the number of sites is overall very limited, and can become insufficient for a robust PCA when further breaking down the dataset. This is why we only included the ‘forest’ subgroup in our original supplementary figures. To address both points raised by the reviewers here (dissecting the dataset, and comparing different results directly), we decided to update our supplementary figures to include side by side results for: 1) all sites (same as in the main text), 2) all forest sites, and 3) only evergreen needle forests (the most abundant site type in our dataset, and the only one with enough sites to run robust PCAs). Accordingly, we modified the wording in the different subsections of the results and methods to refer to the updated supplementary results, as shown below:

“Restricting the analysis to forest sites, or evergreen needleleaf forest sites, produced similar results on PC1 as for the overall case with all sites. This hints at the importance of the leaf economics spectrum both within and across vegetation classes (Supplementary Fig. 1, Supplementary Table 2).” (L175-178)

“In the results based on the forest sites, the number of retained principal components was two, and only one when considering exclusively evergreen needleleaf forest sites. In these subcases,

the plane between the performance-persistence trade, and the size axis, was less pronounced (Supplementary Fig. 3, Supplementary Table 3). (L270-273)

“Analysis of forest sites, and evergreen needleleaf forest sites, produced similar results as for the overall case with all sites. The directionality of the relationships between variables were similar to the overall results, albeit less pronounced (Supplementary Fig. 8, Supplementary Table 4).” (L364-367)

“For all analyses, we repeated the test for 1) all available sites in our dataset, 2) forest sites only, and 3) evergreen needleleaf forest sites only.” (L654-656)

See below also the updated figures with the corresponding captions.

Supplementary Fig. 1. Principal Component Analysis (PCA) on variables representing the Leaf Economics Spectrum at the ecosystem scale on all sites (a, d, g, j, 90 sites), forest sites (b, e, h, k, 69 sites), and only evergreen needleleaf forests (c, f, i, l, 45 sites). For better comparison, we kept the same number of principal components as in the main analysis with all sites. **a, b, c)** Biplot resulting from PCA; point colours represent plant functional types following the IGBP

classification: CSH (Closed Shrubland), DBF (Deciduous Broadleaf Forest), EBF (Evergreen Broadleaf Forest), ENF (Evergreen Needleleaf Forest), GRA (Grassland), MF (Mixed Forest), OSH (Open Shrubland), SAV (Savannah), WET (Wetland), WSA (Woody Savannah). Bigger points represent the centroid of the distribution for each habitat type. **d, e, f**) Explained variance for the retained Principal Components (PCs). **g, h, i**) Barplot for the loadings, and **j, k, l**) contributions for each variable on the retained PCs. Error bars represent the standard error estimated with bootstrap procedure ($n = 499$). Variable acronyms: photosynthetic capacity at light saturation (GPPsat), maximum ecosystem respiration (RECOMax), community-weighted mean leaf longevity (wLL), community-weighted mean leaf mass per area (wLMA), community-weighted mean nitrogen per leaf mass (wNmass).

Supplementary Fig. 3. Principal Component Analysis on the Global Spectrum of Plant Form and Function at the ecosystem scale on all sites (a, d, g, j, 89 sites), forest sites (b, e, h, k, 71 sites), and only evergreen needleleaf forests (c, f, i, l, 46 sites). For better comparison, we kept the same number of principal components as in the main analysis with all sites. **a, b, c**) Biplot resulting from PCA; point colours represent plant functional types following the IGBP classification: CSH (Closed Shrubland), DBF (Deciduous Broadleaf Forest), EBF (Evergreen Broadleaf Forest), ENF (Evergreen Needleleaf Forest), GRA (Grassland), MF (Mixed Forest), OSH (Open Shrubland), SAV (Savannah), WET (Wetland), WSA (Woody Savannah). Bigger points represent the centroid of the distribution for each habitat type. **d, e, f**) Explained variance for the retained Principal Components (PCs). **g, h, i**) Barplot for the loadings, and **j, k, l**) contributions for each variable on the retained PCs. Error bars represent the standard error estimated with bootstrap procedure ($n = 499$). Variable acronyms: photosynthetic capacity at light saturation (GPPsat), maximum ecosystem respiration (RECOMax), community-weighted mean leaf longevity (wLL), community-weighted mean leaf mass per area (wLMA), community-weighted mean nitrogen per leaf mass (wNmass).

(Evergreen Needleleaf Forest), GRA (Grassland), MF (Mixed Forest), OSH (Open Shrubland), SAV (Savannah), WET (Wetland), WSA (Woody Savannah). Bigger points represent the centroid of the distribution for each habitat type. **d, e, f**) Explained variance for the retained Principal Components (PCs). **g, h, i**) Barplot for the loadings, and **j, k, l**) contributions for each variable on the retained PCs. Error bars represent the standard error estimated with bootstrap procedure ($n = 499$). Variable acronyms: gross primary productivity at light saturation (GPPsat), canopy height (Hc), maximum leaf area index (LAI_{max}), community-weighted mean leaf mass per area (wLMA), community-weighted mean nitrogen per leaf mass (wNmass), community-weighted mean stem specific density (wSSD).

Supplementary Fig. 8. Principal Component Analysis (PCA) on the Least-Cost Hypothesis at the ecosystem scale on all sites (a, d, g, j, 82 sites), forest sites (b, e, h, k, 60 sites), and only evergreen needleleaf forests (c, f, i, l, 41 sites). For better comparison, we kept the same number of principal components as in the main analysis with all sites. **a, b, c**) Biplot resulting from PCA; point colours represent plant functional types following the IGBP classification: CSH (Closed Shrubland), DBF (Deciduous Broadleaf Forest), EBF (Evergreen Broadleaf Forest), ENF (Evergreen Needleleaf Forest), GRA (Grassland), MF (Mixed Forest), OSH (Open Shrubland), SAV (Savannah),

WET (Wetland), WSA (Woody Savannah). Bigger points represent the centroid of the distribution for each habitat type. **d, e, f**) Explained variance for the retained Principal Components (PCs). **g, h, i**) Barplot for the loadings, and **j, k, l**) contributions for each variable on the retained PCs. Error bars represent the standard error estimated with bootstrap procedure (n = 499). Variable acronyms: evaporative fraction (EF), maximum surface conductance (G_{max}), canopy height (H_c), photosynthetic nitrogen use efficiency (PNUE), air temperature (T_a), water use efficiency based on transpiration (WUE_t).

Finally, we suggest that the authors make the dataset available, which would obviously simplify the further use of this work for model development.

We made the data and the code available (L667-683):

Data Availability

Data used for this study are the FLUXNET dataset LaThuile (<https://fluxnet.fluxdata.org/data/la-thuille-dataset/>) and FLUXNET2015 (<https://fluxnet.fluxdata.org/data/fluxnet2015-dataset/>). Biological, Ancillary, Disturbance and Metadata for the sites were collected from databases and literature. Plant traits measurements were collected from the TRY database (<https://www.try-db.org/TryWeb/Home.php>). All input data used for the analysis are available on zenodo at <https://doi.org/10.5281/zenodo.7782252>.

Code availability

*All the analyses were conducted with R 4.1.0 for Windows (64-bit). The R packages used for the calculation of the ecosystem functional properties are already described in the literature and freely available on CRAN and git: REddyProc v1.2.2 (Wutzler et al., 2018, <https://cran.rproject.org/web/packages/REddyProc/index.html>), *bingleaf* v0.8.2 (<https://cran.r-project.org/web/packages/bingleaf/index.html>). The R code used for the statistical analyses uses freely available packages described in the Methods section: FactoMineR v2.6, ade4 v1.7-20, modelr v0.1.9, MuMIn v1.43.17, relaimpo v2.2-6. The TEA algorithm is available at*

<https://doi.org/10.5281/zenodo.3921923>. The R codes used for this analysis are available on zenodo at <https://doi.org/10.5281/zenodo.7782252>.

Oskar Franklin & Florian Hofhansl, IIASA

Reference

Franklin, O., Harrison, S.P., Dewar, R., Farrior, C.E., Brännström, Å., Dieckmann, U. et al. (2020). Organizing principles for vegetation dynamics. *Nat. Plants*, 6, 444-453.

References

1. Díaz, S. et al. The global spectrum of plant form and function. *Nature* **529**, 167–171 (2016).

REVIEWERS' COMMENTS

Reviewer #1 (Remarks to the Author):

I appreciate the efforts from the authors. The manuscript has been greatly improved. I echo the comment from reviewer 2 that dividing the manuscript into results, discussion and conclusion will greatly improve the readability, in particular for this study with many assumption-makings from other literatures. I do not think it will clash with the requirement from nature communications(<https://www.nature.com/ncomms/submit/article>), however, that is up to the discretion of the editor. Additionally, I think the manuscript could benefit from a few more rounds of proofreading as I still find some inconsistent or ambiguous statements. Please see some comments below:

L80-81. It seems you are implying that vegetation properties and ecosystem level plant traits were not acquired on flux sites?

L82. Are you suggesting that only leaf-scale coordination is conserved at the ecosystem scale, but plant-scale coordination (as you mentioned in L78) is not conserved?

L83. While I understand there is a tight word limits for abstract, I nevertheless find it is necessary to include some basic information – rough description such as “additional processes” just make readers lost on what processes are you referring to.

L123. “light response curves observed at the leaf level can be very different from the ones observed at the canopy scale” – not sure if the statement is correct, the general shape of light response curve is pretty similar across the scales if we look at the same time scale. In fact, that is the reason there is a daytime partition method in current flux data production, where a light response curve is fitted to daytime carbon flux data to estimate GPP (Lasslop 2010, GCB).

L122 – 124. I find the newly added statement no convincing – I am not sure the references 19 and 20 support your argument here. Ref 19 discussed the role of diffuse radiation on photosynthesis, Ref 20 discussed the how shaded leaves are different than sunlit leaves, not sure the optimality theory is discussed there.

L177. Vegetation classes mean “plant functional types” – please be consistent in using terminologies.

L179 – 182. I still feel this does not provide a good reason to use Nmass instead of Narea, though I am happy to see the authors have also tested Narea. Just note in dynamic vegetation models Narea is a more commonly used parameter, and then we simulate photosynthesis, then biomass and then LMA, so you can see how they are interdependent on each other. As I mentioned, it is expected that Nmass and LMA give you better PCA result since they are correlated and enhance the variance explained along the PC1 dimension. Along with L194-199, those should be in the Discussion.

L213. Here it is mentioned 90 sites were used, but in the abstract there are 98 sites. It seems different number of sites are used in each section of analysis, so I suggest not to emphasize you used 98 sites in the abstract (because most analysis did not).

L231. Though in the rebuttal letter the authors said they use Dray method is all three section of the analysis, here in section 2 it is the first/only time I saw this method is mentioned.

L333. It would be helpful to clarify the meaning of least cost hypothesis here and what are we expected to see if the least cost hypothesis is conserved at the ecosystem scale.

L342-349. So you are suggesting the PC1 in this does not reflect least cost hypothesis?

Reviewer #2 (Remarks to the Author):

We find that most of our concerns have been addressed. In particular, there are now much better explanations of the value for DGVM development. We only have some suggestions for minor adjustments:

Line 76 in the abstract states “However, it is unclear whether tradeoffs and optimality principles in functional traits of leaves are conserved at the ecosystem level.” Many readers would be confused by this statement. We suggest complementing this sentence with “..., i.e. if similar trade-offs are observed between corresponding community mean traits and process at the ecosystem level.

Next sentence line 79 says “Here, we tested three well-known leaf- and plant-level coordination theories at the ecosystem scale: the leaf economics spectrum, the global spectrum of plant form and function, and the least-cost hypothesis.”. But you didn’t test these theories because they apply only on the plant level, you tested the hypothesis that the trait correlation patterns generated by these principles are observed also among community mean traits and ecosystem processes.

Line 81 “ecosystem-level plant traits” is better called community mean traits.

Line 512 says “Furthermore, dynamic global vegetation models should be tested with and without optimality included.” Why should they be tested without optimality?

Line 516 says “The validation of established optimality principles at different scales would support a more accurate implementation of leaf-level theories in models.”. This again is confusing. The leaf level theories used in the models can only be validated with leaf level data. Your results support: 1) the benchmarking, see above, and 2) The possibility to use ecosystem level correlation-principles to make simpler DGVMs that skip over the individual scale processes and model ecosystem processes directly.

Reviewer #1 (Remarks to the Author):

Dear reviewer,

Thank you for your time and constructive comments. We have included changes where appropriate. In cases where suggested changes were not made, we tried to explain why.

In particular, we have addressed the Narea-Nmass debate by including the results based on Narea in the Supplementary Information.

Below we report the point-by-point replies to the comments. The reviewer's comments are reported in italics and grayed-out, our response is in normal text, and the parts added or changed in the manuscript are reported between quotes, with corresponding line numbers.

I appreciate the efforts from the authors. The manuscript has been greatly improved. I echo the comment from reviewer 2 that dividing the manuscript into results, discussion and conclusion will greatly improve the readability, in particular for this study with many assumption-makings from other literatures. I do not think it will clash with the requirement from nature communications (<https://www.nature.com/ncomms/submit/article>), however, that is up to the discretion of the editor. Additionally, I think the manuscript could benefit from a few more rounds of proofreading as I still find some inconsistent or ambiguous statements. Please see some comments below:

L80-81. It seems you are implying that vegetation properties and ecosystem level plant traits were not acquired on flux sites?

That is partly correct. In the methods, we explain what measurements are site-specific, and what plant traits had to be acquired from different sources. In the abstract, this explanation would take up too much word count and is thus only implied.

L82. Are you suggesting that only leaf-scale coordination is conserved at the ecosystem scale, but plant-scale coordination (as you mentioned in L78) is not conserved?

No, this was not our intent. In the reworked abstract following the editor's guidelines, this formulation is not present anymore, which should avoid potential confusion.

L83. While I understand there is a tight word limits for abstract, I nevertheless find it is necessary to include some basic information – rough description such as “additional processes” just make readers lost on what processes are you referring to.

Following the editor's suggestion, we removed the vague formulation of "additional processes", and only mentioned that "we also find evidence of additional scale-emergent properties". We understand that this does not entirely satisfy the need for a more in-depth explanation of what is meant, but doing so would take up too much space in the abstract, and is instead described in length in the introduction and discussion sections.

L123. "light response curves observed at the leaf level can be very different from the ones observed at the canopy scale" – not sure if the statement is correct, the general shape of light response curve is pretty similar across the scales if we look at the same time scale. In fact, that is the reason there is a daytime partition method in current flux data production, where a light response curve is fitted to daytime carbon flux data to estimate GPP (Lasslop 2010, GCB).

We changed this formulation to: "...whereas light-use efficiency responses observed at the leaf level depend on rather homogenous small-scale conditions, complex gradients of light penetration and light-use efficiencies need to be considered at the canopy scale".

L122 – 124. I find the newly added statement no convincing – I am not sure the references 19 and 20 support your argument here. Ref 19 discussed the role of diffuse radiation on photosynthesis, Ref 20 discussed the how shaded leaves are different than sunlit leaves, not sure the optimality theory is discussed there.

We changed this paragraph to be more clear and better match with the references: "For instance, light interception is largely dependent on canopy architecture due to the amount of light that can penetrate the canopy space^{19,20}: whereas light-use efficiency responses observed at the leaf level depend on rather homogenous small-scale conditions, complex gradients of light penetration and light-use efficiencies need to be considered at the canopy scale²¹. In essence, the coordination between ecosystem functional properties at the canopy scale can contrast with the theory of optimization in leaves or plant organs." (L123-129).

Here the references are:

19. Knohl, A. & Baldocchi, D. D. Effects of diffuse radiation on canopy gas exchange processes in a forest ecosystem. *Journal of Geophysical Research: Biogeosciences* 113, (2008).
20. Keenan, T. F. & Niinemets, Ü. Global leaf trait estimates biased due to plasticity in the shade. *Nature Plants* 3, 1–6 (2016).
21. Baldocchi, D. D. Measuring fluxes of trace gases and energy between ecosystems and the atmosphere – the state and future of the eddy covariance method. *Global Change Biology* 20, 3600–3609 (2014).

L177. Vegetation classes mean "plant functional types" – please be consistent in using terminologies.

Thank you for spotting this inconsistency in the terminology. We changed it to be consistent throughout the text.

L179 – 182. I still feel this does not provide a good reason to use Nmass instead of Narea, though I am happy to see the authors have also tested Narea. Just note in dynamic vegetation models Narea is a more commonly used parameter, and then we simulate photosynthesis, then biomass and then LMA, so you can see how they are interdependent on each other. As I mentioned, it is expected that Nmass and LMA give you better PCA result since they are correlated and enhance the variance explained along the PC1 dimension. Along with L194-199, those should be in the Discussion.

Given that both approaches come with advantages and disadvantages, we decided to include the plot with wNarea in the Supplementary Information, and referenced it in the discussion: “We also tested the same concept with area-based nitrogen estimates, and we observed very similar results (Supplementary Fig. 3, Supplementary Data 1)” (L202-203).

L213. Here it is mentioned 90 sites were used, but in the abstract there are 98 sites. It seems different number of sites are used in each section of analysis, so I suggest not to emphasize you used 98 sites in the abstract (because most analysis did not).

Following this comment and the editor’s suggestion, we modified the abstract to specify the number of sites for the individual analyses (90 for the leaf economics spectrum, 89 for the global spectrum, and 82 for the least cost hypothesis).

L231. Though in the rebuttal letter the authors said they use Dray method in all three section of the analysis, here in section 2 it is the first/only time I saw this method is mentioned.

We only mention the Dray method in the second part because it’s the only time that we “ignore” its results of six retained PCs and only focus on three. In both other cases (leaf economics spectrum and least cost hypothesis) we stick to the number of retained axes selected by the Dray method, which is described in the Methods section: “For each section of the analysis, we tested the number of significant principal components to be retained following Dray’s method²⁵, using the ade4 R package^{64,65}, in order to minimize redundancy as well as loss of information.” (L613-616).

L333. It would be helpful to clarify the meaning of least cost hypothesis here and what are we expected to see if the least cost hypothesis is conserved at the ecosystem scale.

We expanded the beginning of the results section for the least cost hypothesis by adding: “For the analyses of the least-cost hypothesis^{7,11} at the ecosystem scale we focused primarily on the expected trade-off between the costs in the acquisition, retention, and use-efficiencies of nitrogen and water. Therefore, we considered variables directly or indirectly related to the costs of nitrogen [...] and water...” (L311-314)

L342-349. So you are suggesting the PC1 in this does not reflect least cost hypothesis?

Yes, we think that the trade-offs related to the least-cost hypothesis more strongly emerge on PC2, as we state in the highlighted section: “The second principal component [...] uncovered the

trade-offs expected by the least-cost hypothesis...” (L331-332). We also state this more clearly in the discussion: “...the least-cost hypothesis [is] also evident for whole ecosystems, despite embodying secondary mechanisms at the ecosystem scale.” (L422-424).

Reviewer #2 (Remarks to the Author):

Dear reviewer,

Thank you for your time and constructive comments. We have included changes where appropriate. In cases where suggested changes were not made, we tried to explain why.

Below we report the point-by-point replies to the comments. The reviewer's comments are reported in italics and grayed-out, our response is in normal text, and the parts added or changed in the manuscript are reported between quotes, with corresponding line numbers.

We find that most of our concerns have been addressed. In particular, there are now much better explanations of the value for DGVM development. We only have some suggestions for minor adjustments:

Line 76 in the abstract states “However, it is unclear whether tradeoffs and optimality principles in functional traits of leaves are conserved at the ecosystem level.” Many readers would be confused by this statement. We suggest complementing this sentence with “..., i.e. if similar tradeoffs are observed between corresponding community mean traits and process at the ecosystem level.

Based on this comment and the suggestions from the editor, we changed the corresponding lines in the abstract to: “Here, we test whether trait correlation patterns predicted by three well-known leaf- and plant-level coordination theories – the leaf economics spectrum, the global spectrum of plant form and function, and the least-cost hypothesis – are also observed between community mean traits and ecosystem processes.” (L79-82).

Next sentence line 79 says “Here, we tested three well-known leaf- and plant-level coordination theories at the ecosystem scale: the leaf economics spectrum, the global spectrum of plant form and function, and the least-cost hypothesis.”. But you didn’t test these theories because they apply only on the plant level, you tested the hypothesis that the trait correlation patterns generated by these principles are observed also among community mean traits and ecosystem processes.

Combining this suggestion with the previous comment, we change the corresponding part in the abstract to: “Here, we test whether trait correlation patterns predicted by three well-known leaf- and plant-level coordination theories – the leaf economics spectrum, the global spectrum of plant form and function, and the least-cost hypothesis – are also observed between community mean traits and ecosystem processes.” (L79-82).

Line 81 “ecosystem-level plant traits” is better called community mean traits.

We changed the corresponding formulation with “community mean plant traits”, as suggested.

Line 512 says “Furthermore, dynamic global vegetation models should be tested with and without optimality included.” Why should they be tested without optimality?

The comparison between models with and without optimality should help to understand if and how much the inclusion of optimality actually affects the predictions.

Line 516 says “The validation of established optimality principles at different scales would support a more accurate implementation of leaf-level theories in models.”. This again is confusing. The leaf level theories used in the models can only be validated with leaf level data. Your results support: 1) the benchmarking, see above, and 2) The possibility to use ecosystem level correlation-principles to make simpler DGVMs that skip over the individual scale processes and model ecosystem processes directly.

We changed the last sentence to be less confusing: “The validation of established optimality principles at different scales would support a more accurate implementation of the notions learned from leaf-level theories in models across scales.” (L486-488).